# NODE-TIME CONDITIONAL PROMPT LEARNING IN DYNAMIC GRAPHS

**Xingtong Yu**[1*], **Zhenghao Liu**[2*], **Xinming Zhang**[2†], **Yuan Fang**[1†]

Singapore Management University[1], University of Science and Technology of China[2]

{xingtongyu, yfang}@smu.edu.sg, salzh@mail.ustc.edu.cn, xinming@ustc.edu.cn

## ABSTRACT

Dynamic graphs capture evolving interactions between entities, such as in social networks, online learning platforms, and crowdsourcing projects. For dynamic graph modeling, dynamic graph neural networks (DGNNs) have emerged as a mainstream technique. However, they are generally pre-trained on the link prediction task, leaving a significant gap from the objectives of downstream tasks such as node classification. To bridge the gap, prompt-based learning has gained traction on graphs, but most existing efforts focus on static graphs and neglect the evolution of dynamic graphs. In this paper, we propose DYGPROMPT, a novel pre-training and prompt learning framework for dynamic graph modeling. First, we design *dual prompts* to address the discrepancy in both task objectives and temporal variations across pre-training and downstream tasks. Second, we recognize that node and time patterns often characterize each other, and propose *dual condition-nets* to model the evolving node-time patterns in downstream tasks. Finally, we thoroughly evaluate and analyze DYGPROMPT through extensive experiments on four public datasets.

## 1 INTRODUCTION

Graph data are pervasive due to their ability to model complex relationships between entities in various applications, including social network analysis (Fan et al., 2019; Yu et al., 2023c), Web mining (Xu et al., 2022; Wang et al., 2023), and content recommendation systems (Qu et al., 2023; Zhang et al., 2024). In many of these applications, the graph structures evolve over time, such as users commenting on posts in a social network (Kumar et al., 2018; Iba et al., 2010), editing pages in a crowdsourcing project like Wikipedia (Kumar et al., 2015), or interacting with courses in an online learning platform (Liyanagunawardena et al., 2013). Such graphs are termed *dynamic* graphs (Barros et al., 2021; Skarding et al., 2021), in contrast to conventional *static* graphs that maintain an unchanging structure.

On dynamic graphs, dynamic graph neural networks (DGNNs) (Pareja et al., 2020; Xu et al., 2020) have been widely applied. In a typical design for DGNNs, each node updates its representation by iteratively receiving and aggregating messages from its neighbors in a time-dependent manner. While DGNNs are often pre-trained on the link prediction task, the downstream application could involve a different task, such as node classification, leading to a significant gap between pre-training and task objectives. More advanced pre-training and fine-tuning strategies on dynamic graphs (Bei et al., 2024; Chen et al., 2022; Tian et al., 2021) also suffer from the same limitation. They first pre-train a model for dynamic graphs based on various task-agnostic self-supervised signals on the graph, and then update the pre-trained model weights in a fine-tuning phase based on task-specific labels pertinent to the downstream application. Likewise, the pre-training and downstream objectives can differ significantly, resulting in a notable gap between them.

To bridge the gap between pre-training and downstream objectives (Liu et al., 2023a), prompt learning has first emerged in language models (Brown et al., 2020). Fundamentally, a prompt serves to reformulate the input for the downstream task to align with the pre-trained model, while freezing

---

*Co-first authors.

†Corresponding authors.

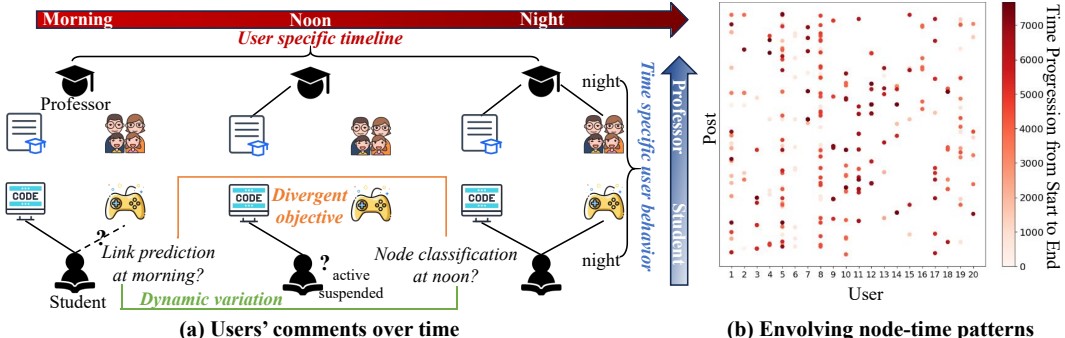

Figure 1: Motivation of DYGPROMPT. (a) Users comment on different topics over time. (b) Evolving node-time patterns in Wikipedia, as node and time mutually characterize each other.

the pre-trained weights. Given far fewer parameters in a prompt than the pre-trained model, prompt learning is parameter-efficient and effective especially in data-scarce scenarios (Yu et al., 2024a). Inspired by the success of prompts in the language domain, recent studies have explored prompt learning on graphs (Liu et al., 2023b; Sun et al., 2022b; 2023). However, these studies only focus on static graphs, neglecting the unique challenges brought by dynamic graphs. In this work, we explore **Prompt** learning for **Dy**namic **G**raphs, and propose a framework called DYGPROMPT. The problem is non-trivial due to two key challenges.

First, how do we design prompts to *bridge temporal variations across time, in addition to divergent task objectives?* Generally, both task objectives and the timing of pre-training and downstream tasks can be inconsistent. For example, consider a user discussion network such as Reddit.com, as illustrated in Fig. 1(a), where users and posts are represented as nodes, and an edge is formed when a user comments on a post. On this dynamic graph, a typical pre-training task involves link prediction—predicting whether a user comments on (i.e., links to) a video game post at time $t_1$. In contrast, the downstream task could be the status classification of the user's account, e.g., determining whether it has been suspended at time $t_2$. While previous work on static graphs (Liu et al., 2023b; Sun et al., 2022b; 2023) employs prompts to narrow the task objective gap, the temporal gap remains unsolved. In particular, even when the downstream task also involves link prediction, the evolving graph structure introduces a temporal gap between the pre-training and downstream phases. In DYGPROMPT, we introduce dual prompts, consisting of a *node prompt* and a *time prompt*. On the one hand, a *node prompt* alters the node features to reformulate task input and bridge the task gap, similar to previous work (Liu et al., 2023b; Fang et al., 2024). On the other hand, a *time prompt* adjusts the time features, capturing the temporal evolution of the dynamic graph, thereby narrowing inconsistencies that arise from varying priorities at different times.

Second, how do we capture *evolving patterns across different nodes and time points, driven by the dynamic interplay between them?* Different nodes exhibit unique timelines, evolving distinctively over time as nodes and time mutually influence each other. For example, in the user discussion network, a student may comment on open source code in the morning, while commenting on video games in the evening. However, as shown in Fig. 1(a), a professor may comment on academic policies at noon, while discussing family topics at night. Therefore, node and time patterns mutually characterize each other: the behavior of the same node may be influenced by different times, while the behavior of a node at a specific time may be influenced by the characteristics of the node. Hence, the evolving node-time patterns, as shown in Fig. 1(b), may differ from the patterns observed in the pre-training data. However, previous studies neglect the fine-grained interplay between the node and time patterns downstream. Note that some works (Tan et al., 2023; Wen & Fang, 2023) have designed individual prompts or tokens tailored to each node, and a contemporary approach (Chen et al., 2024) has developed time-aware prompts. However, they do not account for the mutual characterization between nodes and time, failing to capture the fine-grained, evolving node-time patterns. In DYGPROMPT, inspired by conditional prompt learning (Zhou et al., 2022), we propose a series of node and time prompts generated by *dual condition-nets*. Specifically, a *time condition-net* generates a sequence of time-conditioned node prompts for each node. These time-conditioned node prompts reflect temporal subtleties at different times for the same node, thereby better aligning

the node's pattern at different times with those observed during pre-training compared to using a fixed node prompt across all time points. Likewise, a *node condition-net* generates a sequence of node-conditioned time prompts at each timestamp. The node-conditioned time prompts account for node characteristics, thereby better aligning the time patterns of different nodes with those captured by the pre-trained model compared to using a fixed time prompt for all nodes. It is worth noting that the condition-nets generate prompts conditioned on the input features rather than directly parameterizing the prompts, significantly reducing the number of learnable parameters in the downstream prompt-tuning phase.

In summary, the contributions of this work are threefold. (1) We introduce DYGPROMPT for dynamic graph modeling, proposing dual prompts to narrow the gaps arising from both task and dynamic differences between pre-training and downstream phases. (2) We recognize that node and time patterns mutually characterize each other and further propose dual condition-nets to generate a series of node and time prompts, thereby adapting to downstream node-time patterns. (3) We conduct extensive experiments on four benchmark datasets, demonstrating the superior performance of DYGPROMPT in comparison to state-of-the-art approaches.

## 2 RELATED WORK

**Dynamic graph learning.** Real-world graph structures often evolve over time. To model such dynamic evolution, various *continuous-time* dynamic graph learning techniques have been proposed. Typically, these methods update node representations by iteratively receiving and aggregating messages from their neighbors in a time-dependent manner (Skarding et al., 2021; Duan et al., 2024). Some approaches employ dynamic random walks to depict structural changes (Nguyen et al., 2018; Wang et al., 2021), while others incorporate a time encoder to integrate temporal context with structural modeling (Xu et al., 2020; Cong et al., 2022; Rossi et al., 2020; Yu et al., 2023a). Furthermore, some researchers utilize temporal point processes to model structural evolution or node communications (Kumar et al., 2019; Trivedi et al., 2019; Wen & Fang, 2022). However, these methods are typically trained on link prediction tasks, whereas downstream applications may involve different tasks, such as node classification, creating a significant gap between training and task objectives. This gap impedes effective knowledge transfer, adversely impacting downstream task performance.

**Graph pre-training.** More advanced dynamic graph pre-training methods also face similar problems. Extending from graph pre-training methods (Hu et al., 2020a;b; Jiang et al., 2023), dynamic graph pre-training captures inherent properties of dynamic graphs using various task-agnostic, self-supervised signals through strategies such as structural and temporal contrastive learning (Bei et al., 2024; Tian et al., 2021; Li et al., 2022), dynamic graph generation (Chen et al., 2022), and curvature-varying Riemannian graph neural networks (Sun et al., 2022a). They then attempt to transfer the prior knowledge to downstream tasks through fine-tuning with task-specific supervision. Nonetheless, a gap persists between the objectives of pre-training and fine-tuning (Liu et al., 2023a; Yu et al., 2023d). While pre-training seeks to extract fundamental insights from graphs without supervision, fine-tuning is tailored to specific supervision for a given downstream task.

**Graph prompt learning.** First emerging in the language domain, prompt learning effectively bridges the gap between pre-training and downstream objectives (Brown et al., 2020). With a unified template, prompts are specifically tailored for each downstream task, aligning it more closely with the pre-trained model while freezing the pre-trained weights. Prompt learning has been widely adopted on static graphs (Liu et al., 2023b; Sun et al., 2023; Fang et al., 2024; Tan et al., 2023; Yu et al., 2023d; 2025b; 2024b; 2025a;c; 2023b), yet its application in dynamic graphs remains an open question. A contemporary study, TIGPrompt (Chen et al., 2024), proposes a prompt generator to learn time-aware node prompts. However, it only considers the temporal factor in node features, overlooking that time prompts for each node can also be influenced by node features.

## 3 PRELIMINARIES

In this section, we present related preliminaries and introduce our scope.

**Dynamic graph.** Dynamic graphs are categorized into *discrete-time* dynamic graphs and *continuous-time* dynamic graphs (Skarding et al., 2021). The former consists of a series of static graph snapshots, whereas the latter is depicted as a sequence of events on a continuous timeline. In this work, we adopt the more general continuous-time definition, where a *dynamic graph* is represented by $G = (V, E, T)$. Here, $V$ denotes the set of nodes, $E$ denotes the set of edges, and $T$ represents the time domain. In particular, each edge $(v_i, v_j, t) \in E$ indicates a specific interaction (or event) between nodes $v_i$ and $v_j$ at time $t$. Each node is associated with a temporal feature vector $\mathbf{x}_{t,v} \in \mathbb{R}^d$ evolving over time, represented as a row in the temporal feature matrix $\mathbf{X}_t \in \mathbb{R}^{|V| \times d}$.

**Dynamic graph neural network.** A popular backbone for static graphs is message-passing graph neural networks (GNNs) (Wu et al., 2020). Formally, in the $l$-th GNN layer, the embedding of node $v$, represented as $\mathbf{h}_v^l$, is calculated based on the embeddings from the preceding layer:

$$\mathbf{h}_v^l = \texttt{Aggr}(\mathbf{h}_v^{l-1}, \{\mathbf{h}_u^{l-1} : u \in \mathcal{N}_v\}), \tag{1}$$

where $\mathcal{N}_v$ denotes the set of neighboring nodes of $v$, and $\texttt{Aggr}(\cdot)$ is a neighborhood aggregation function. Extending to dynamic graph neural networks (DGNNs), researchers integrate temporal contexts into neighborhood aggregation (Pareja et al., 2020; Xu et al., 2020), often utilizing a *time encoder* (TE) to map timestamps or intervals (Cong et al., 2022; Rossi et al., 2020) within a *dynamic graph encoder* (DGE). Consequently, in the $l$-th layer of a dynamic graph encoder, the embedding of node $v$ at time $t$, denoted by $\mathbf{h}_{t,v}^l$, is derived as

$$\mathbf{h}_{t,v}^l = \texttt{Aggr}(\texttt{Fuse}(\mathbf{h}_{t,v}^{l-1}, \texttt{TE}(t)), \{\texttt{Fuse}(\mathbf{h}_{t',u}^{l-1}, \texttt{TE}(t')) : (u, t') \in \mathcal{N}_v\}), \tag{2}$$

where $\mathcal{N}_v$ contains the historical neighbors of $v$, with $(u, t') \in \mathcal{N}_v$ indicating that $u$ interacts with $v$ at time $t' < t$. $\texttt{Fuse}(\cdot)$ is a fusion operation that integrates structural and temporal contexts, such as concatenation (Rossi et al., 2020) or addition (Trivedi et al., 2019). For simplicity, the output node embedding from the final layer is denoted as $\mathbf{h}_{t,v}$, which is then utilized in a loss function for optimization. Note that the time encoder maps the time domain to a vector space, $\texttt{TE} : T \to \mathbb{R}^d$. A common implementation is $\texttt{TE}(t) = \frac{1}{\sqrt{d}}[\cos(\omega_1 t), \sin(\omega_1 t), \ldots, \cos(\omega_{d/2} t), \sin(\omega_{d/2} t)]$, where $\{\omega_1, \ldots, \omega_{d/2}\}$ is a set of learnable parameters (Xu et al., 2020; Cong et al., 2022).

**Scope of work.** In this work, we pre-train a DGNN with a time encoder on dynamic graphs, employing the task of temporal link prediction (Wang et al., 2021). For downstream tasks, we target two popular dynamic graph-based tasks, namely, temporal node classification and temporal link prediction[1]. Specifically, we focus on *data-scarce scenarios* with only limited data for downstream adaptation, as labeled data are often difficult or costly to obtain for node classification (Zhou et al., 2019; Yao et al., 2020), while nodes with sparse interactions are common in real-world link prediction tasks (Lee et al., 2019; Pan et al., 2019).

## 4 PROPOSED APPROACH

In this section, we present our proposed model DYGPROMPT, starting with its overall framework, followed by its key components.

### 4.1 OVERALL FRAMEWORK

We illustrate the overall framework of DYGPROMPT in Fig. 2, which consists of two phases: pre-training and downstream adaptation. First, given a dynamic graph in Fig. 2(a), we pre-train a DGNN based on temporal link prediction, as shown in Fig. 2(b). In particular, we employ a universal task template based on similarity calculation (Liu et al., 2023b), which unifies different graph-based tasks, such as link prediction and node classification, across the pre-training and downstream phases. Second, given a pre-trained DGNN, to enhance its adaptation to downstream tasks, we propose *dual prompts*, as shown in Fig. 2(c). The dual prompts comprise a *node prompt* and a *time prompt*, aiming to bridge the gaps caused by different task objectives and dynamic variations, respectively. Moreover, extending the dual prompts, we propose *dual condition-nets* to generate a

---

[1]As dynamic graphs focus on the evolving structures within a graph, graph classification is rarely evaluated as a task (Skarding et al., 2021; Pareja et al., 2020; Xu et al., 2020; Chen et al., 2024).

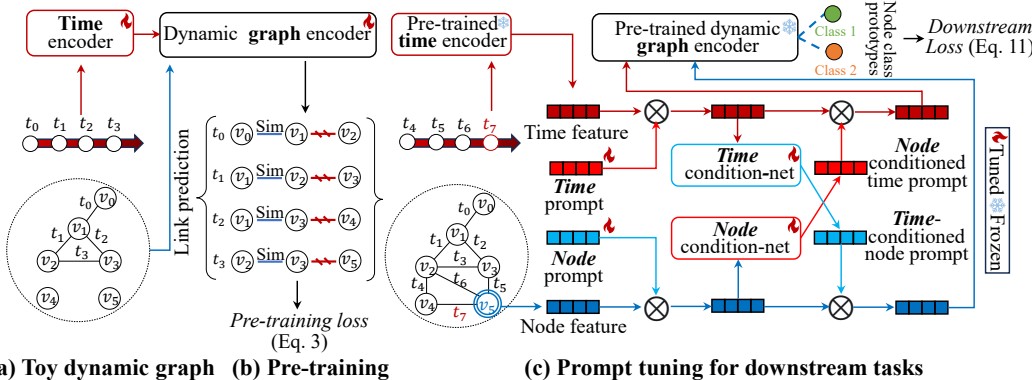

Figure 2: Overall framework of DYGPROMPT.

series of time-conditioned node prompts and node-conditioned time prompts. These conditional prompts are designed to capture the evolving patterns across nodes and time points, along with their mutual characterization, in a fine-grained and parameter-efficient manner.

## 4.2 PRE-TRAINING

Pre-training techniques (Le-Khac et al., 2020) have been widely explored on dynamic graphs (Bei et al., 2024; Chen et al., 2022; Tian et al., 2021). In particular, the most common pre-training task is temporal link prediction (Xu et al., 2020; Rossi et al., 2020), where links are readily available without the need for manual annotations. Following these prior works, we adopt link prediction for pre-training, as illustrated in Fig. 1(b). Its simplicity underscores the effectiveness and robustness of our prompt-based learning in downstream tasks.

Specifically, consider an event $(v, a, t)$, i.e., an interaction between nodes $v$ and $a$ at time $t$. Correspondingly, we construct a tuple $(v, a, b, t)$ such that nodes $v$ and $b$ are not linked at $t$, serving as a contrastive signal. For each tuple, the pre-training objective is to maximize the similarity between nodes $v$ and $a$ while minimizing that between $v$ and $b$. Similar to previous work (You et al., 2020; Yu et al., 2023d), we optimize the following pre-training loss over pre-training data, $\mathcal{D}_{\text{pre}}$.

$$\mathcal{L}_{\text{pre}}(\mathcal{D}_{\text{pre}}; \Theta) = -\sum\nolimits_{(v,a,b,t)\in\mathcal{D}_{\text{pre}}} \ln \frac{\exp\left(\frac{1}{\tau}\text{sim}(\mathbf{h}_{t,v}, \mathbf{h}_{t,a})\right)}{\exp\left(\frac{1}{\tau}\text{sim}(\mathbf{h}_{t,v}, \mathbf{h}_{t,b})\right)}. \tag{3}$$

In Eq. (3), $\mathbf{h}_{t,v}, \mathbf{h}_{t,a}, \mathbf{h}_{t,b}$ represent the node embeddings of $v, a, b$ at time $t$, respectively. $\tau > 0$ is a temperature hyperparameter. $\Theta$ denotes the set of learnable parameters of the DGNN, including those of the dynamic graph encoder and the time encoder. The pre-trained parameters, $\Theta_{\text{pre}} = \arg\min_{\Theta} \mathcal{L}_{\text{pre}}(\mathcal{D}_{\text{pre}}; \Theta)$, will be used to initialize the downstream models.

## 4.3 DUAL PROMPTS

To reconcile task and temporal variations across pre-training and downstream tasks, we propose *dual prompts*, comprising a node prompt and a time prompt, as shown in Fig. 2(c).

**Node prompt.** In language models, a prompt is a purposely designed textual description that reformulates the input for a downstream task, aligning the task with the pre-trained model (Brown et al., 2020; Jia et al., 2021). Likewise, a prompt for static graphs adjusts downstream node features (Liu et al., 2023b; Fang et al., 2024). In line with previous graph prompt learning methods on static graphs (Liu et al., 2023b; Yu et al., 2023d;e; 2025b), for each downstream task, we define a learnable vector $\mathbf{p}^{\text{node}}$ as the node prompt to modify the node features at a given time via element-wise multiplication. Concretely, the input features of node $v$ at time $t$, denoted by $\mathbf{x}_{t,v}$, are modified as:

$$\mathbf{x}_{t,v}^{\text{node}} = \mathbf{p}^{\text{node}} \odot \mathbf{x}_{t,v}, \tag{4}$$

where $\odot$ is element-wise multiplication. The node prompt $\mathbf{p}^{\text{node}}$ has the same dimension as node features, altering the importance of various node feature dimensions.

**Time prompt.** We adopt a time encoder TE to map the continuously valued time $t$ to time features, $\mathbf{f}_t = \text{TE}(t)$. However, TE is optimized based on the time periods in the pre-training data, which can be distinct from those in downstream tasks. To overcome the temporal gap, we propose a time prompt to adjust time features. Formally, for each downstream task, we define a learnable vector $\mathbf{p}^{\text{time}}$ as the *time prompt*. Then, the time feature at $t$, denoted by $\mathbf{f}_t$, is reformulated as follows.

$$\mathbf{f}_t^{\text{time}} = \mathbf{p}^{\text{time}} \odot \mathbf{f}_t. \tag{5}$$

### 4.4 Dual condition-nets

In dynamic graphs, the same node may exhibit distinct behaviors at different times. Meanwhile, different nodes may evolve differently over the same time interval. In other words, nodes and their temporal patterns influence and characterize each other over time, revealing fine-grained node-time patterns. To capture such interplay, we turn to conditional prompt learning (Zhou et al., 2022). Instead of learning a static prompt tailored to a specific task over the entire time span, we design *dual condition-nets*, comprising a time condition-net that generates a sequence of time-conditioned node prompts for each node and a node condition-net that generates a sequence of node-conditioned time prompts at each timestamp, as shown in Fig. 2(c). In particular, each condition-net utilizes a lightweight network to generate prompts conditioned on the input features, enabling a more parameter-efficient approach to prompt tuning compared to directly parameterizing the prompts for each node and timestamp.

**Time condition-net.** To incorporate time characteristics into node prompts, a straightforward way is to employ a series of learnable vectors as prompts for each distinct timestamp. However, this approach does not scale to large time intervals with many timestamps, as it quickly increases the number of tunable parameters. Therefore, we propose a *time condition-net*, which generates a series of *time-conditioned node prompts*. Formally, the node prompt conditioned on time $t$ is given by

$$\tilde{\mathbf{p}}_t^{\text{node}} = \text{TCN}(\mathbf{f}_t^{\text{time}}; \kappa), \tag{6}$$

where TCN is the time condition-net parameterized by $\kappa$. This can be viewed as a form of hypernetwork (Ha et al., 2022), where our condition-net TCN serves as a secondary network to generate the node prompts conditioned on the input time features. This formulation enables parameter-efficient prompt generation, especially when using a lightweight condition-net. In our implementation, we opt for a simple multi-layer perceptron (MLP) with an efficient bottleneck structure (Wu & Lee, 2018). Therefore, for a series of events occurring at times $\{t_0, t_1, \ldots\}$, the time condition-net produces their corresponding time-conditioned node prompts $\{\tilde{\mathbf{p}}_{t_0}^{\text{node}}, \tilde{\mathbf{p}}_{t_1}^{\text{node}}, \ldots\}$. Note that these conditional prompts match the dimensionality of the node features. Then, the features of node $v$ at time $t$, which have been modified by the node prompt in Sect. 4.3, are further updated as

$$\tilde{\mathbf{x}}_{t,v}^{\text{node}} = \tilde{\mathbf{p}}_t^{\text{node}} \odot \mathbf{x}_{t,v}^{\text{node}}. \tag{7}$$

**Node condition-net.** Similarly, to incorporate node characteristics into time prompts, we propose a *node condition-net* to generate a series of *node-conditioned time prompts*, mirroring the time condition-net. The time prompt conditioned on the node features of $v$ at time $t$ is given by

$$\tilde{\mathbf{p}}_{t,v}^{\text{time}} = \text{NCN}(\mathbf{x}_{t,v}^{\text{node}}; \phi), \tag{8}$$

where NCN is the node condition-net parameterized by $\phi$, which is also implemented as an MLP with a bottleneck structure. Consequently, at time $t$, a series of node-conditioned time prompts $\{\tilde{\mathbf{p}}_{t,v_0}^{\text{time}}, \tilde{\mathbf{p}}_{t,v_1}^{\text{time}}, \ldots\}$ is generated for all nodes, which have the same dimension as the time features. These node-specific time prompts are then applied to each node at time $t$, as follows.

$$\tilde{\mathbf{f}}_{t,v}^{\text{time}} = \tilde{\mathbf{p}}_{t,v}^{\text{time}} \odot \mathbf{f}_t^{\text{time}}. \tag{9}$$

### 4.5 Prompt tuning

Finally, the features adjusted by the prompts are fed into the dynamic graph encoder, DGE. The embedding of node $v$ at time $t$ is calculated as follows:

$$\mathbf{h}_{t,v} = \text{DGE}\left(\text{Fuse}(\tilde{\mathbf{x}}_{t,v}^{\text{node}}, \tilde{\mathbf{f}}_{t,v}^{\text{time}}), \left\{\text{Fuse}(\tilde{\mathbf{x}}_{t',u}^{\text{node}}, \tilde{\mathbf{f}}_{t',v}^{\text{time}}) : (u,t') \in \mathcal{N}_v\right\}\right). \tag{10}$$

To tune the prompts and condition-nets for a downstream task, we adopt similarity calculation as the task template (Liu et al., 2023b), consistent with the pre-training loss. Specifically, for temporal node classification, consider a labeled set $\mathcal{D}_{\text{down}} = \{(v_1, y_1, t_1), (v_2, y_2, t_2), \dots\}$, where each $v_i$ denotes a node, and $y_i \in Y$ is the class label of $v_i$ at time $t_i$. Then, the downstream loss is

$$\mathcal{L}_{\text{down}}(\mathcal{D}_{\text{down}}; \mathbf{p}^{\text{node}}, \mathbf{p}^{\text{time}}, \kappa, \phi) = -\sum_{(v_i, y_i, t_i) \in \mathcal{D}_{\text{down}}} \ln \frac{\exp\left(\frac{1}{\tau} \text{sim}(\mathbf{h}_{t_i, v_i}, \bar{\mathbf{h}}_{t_i, y_i})\right)}{\sum_{y \in Y} \exp\left(\frac{1}{\tau} \text{sim}(\mathbf{h}_{t_i, v_i}, \bar{\mathbf{h}}_{t_i, y})\right)}, \quad (11)$$

where $\bar{\mathbf{h}}_{t_i, y}$ represents class $y$'s prototype embeddings (Liu et al., 2023b) at time $t_i$, which is obtained by the mean embeddings of examples in class $y$ at time $t_i$. For temporal link prediction, we utilize the same loss as the pre-training loss in Eq. (3).

During prompt tuning, we only update the lightweight dual prompt vectors ($\mathbf{p}^{\text{node}}, \mathbf{p}^{\text{time}}$) and the parameters of the dual condition-nets ($\kappa, \phi$), whilst freezing the pre-trained DGNN weights. This parameter-efficient tuning approach is amenable to label-scarce settings, where $\mathcal{D}_{\text{down}}$ comprises only a limited number of training examples. We outline the key steps for prompt tuning in Algorithm 1 in Appendix A and assess its complexity in Appendix B.

## 5 EXPERIMENTS

In this section, we conduct experiments to evaluate DYGPROMPT and analyze the empirical results.

### 5.1 EXPERIMENTAL SETUP

**Datasets.** We utilize four benchmark datasets for evaluation: *Wikipedia*, *Reddit*, *MOOC* and *Genre*. We provide further details of the datasets in Appendix C.

**Downstream tasks.** We conduct temporal node classification and temporal link prediction to evaluate the performance of DYGPROMPT. For link prediction, we perform experiments in both transductive and inductive settings, depending on whether nodes in the testing set have appeared in the pre-training or downstream tuning data. In this study, we investigate the pre-training and prompt learning on dynamic graphs, which differs from previous DGNNs (Rossi et al., 2020; Xu et al., 2020). Therefore, we adopt a new data split to accommodate the pre-training and downstream tuning steps. Specifically, given a *chronologically ordered* sequence of events (i.e., edges with timestamps), we use the first 80% for pre-training. Note that we pre-train a DGNN only once for each dataset and subsequently employ the same pre-trained model for all downstream tasks. The remaining 20% of the events are used for downstream tasks. Specifically, they are further split into 1%/1%/18% subsets, where the first 1% is reserved as the training pool for downstream prompt tuning, and the next 1% as the validation pool, with the last 18% for downstream testing.

To construct a downstream task, we start by sampling 30 events (about 0.01% of the whole dataset) from the training pool, ensuring that there is at least one user per class. Then, for node classification, we take the user nodes as training instances for downstream adaptation, and their ground-truth labels are given by their labels at the time of the corresponding event. For link prediction, the downstream training pool also helps with adaptation due to its temporal proximity to the test set. Specifically, we take the sampled events as positive instances, while further sampling one negative instance for each positive instance. Specifically, for each event $(v, a, t)$, we sample one negative instance $(v, b, t)$ such that $b$ is a node taken from the training pool that is not linked to $v$ at time $t$. The validation sets are constructed by repeating the same process on the validation pool. Finally, the test set for node classification simply comprises all user nodes in the testing events, while the test set for link prediction is expanded to include negative instances following a similar sampling process as in training. Furthermore, for inductive link prediction, we remove instances from the test set if their nodes have appeared in the pre-training or downstream training data. We repeat this sampling process for 100 times to construct 100 tasks for node classification and link prediction.

**Evaluation.** To evaluate the performance on the tasks, we employ AUC-ROC for both node classification (Xu et al., 2020; Rossi et al., 2020) and link prediction (Sun et al., 2022a; Bei et al., 2024), following prior work. For each of the 100 tasks, we repeat the experiments with five different random seeds, and report the average and standard deviation over the 500 results.

Table 1: AUC-ROC (%) evaluation of temporal node classification and link prediction.

| Methods | Node Classification | | | | Transductive Link Prediction | | | | Inductive Link Prediction | | | |
|---|---|---|---|---|---|---|---|---|---|---|---|---|
| | Wikipedia | Reddit | MOOC | Genre | Wikipedia | Reddit | MOOC | Genre | Wikipedia | Reddit | MOOC | Genre |
| GCN-ROLAND | 58.86±10.3 | 48.25±9.57 | 49.93±6.74 | 46.33±3.97 | 49.61±3.12 | 50.01±2.53 | 49.82±1.44 | 49.15±3.74 | 49.60±2.37 | 49.90±1.64 | 49.16±2.48 | 47.25±2.97 |
| GAT-ROLAND | 62.81±9.88 | 47.95±8.42 | 50.01±6.34 | 47.26±3.49 | 52.34±1.82 | 50.04±1.98 | 55.74±3.71 | 47.69±2.81 | 52.29±1.97 | 49.85±2.35 | 54.01±2.16 | 49.38±2.72 |
| TGAT | 67.00±5.35 | 53.64±5.50 | 59.27±4.43 | 51.26±2.31 | 55.78±2.03 | 62.43±1.86 | 51.49±1.30 | 69.11±3.89 | 48.21±1.55 | 57.30±0.70 | 51.42±4.27 | 48.38±4.72 |
| TGN | 50.61±13.6 | 49.54±6.23 | 50.33±4.47 | 50.72±2.31 | 72.48±0.19 | 67.37±0.07 | 54.60±0.80 | 86.46±2.84 | 74.38±0.29 | 69.81±0.08 | 54.62±0.72 | 87.17±2.68 |
| TREND | 69.92±9.27 | 64.85±4.71 | 66.79±5.44 | 50.34±1.62 | 63.24±0.71 | 80.42±0.45 | 58.70±0.78 | 52.78±1.14 | 50.15±0.90 | 65.13±0.54 | 57.52±1.01 | 45.31±0.43 |
| GRAPHMIXER | 65.43±4.21 | 60.21±5.36 | 63.72±4.98 | 50.15±1.49 | 59.73±0.35 | 61.88±0.11 | 52.42±1.38 | 60.83±3.25 | 51.34±0.84 | 57.64±0.31 | 51.16±2.59 | 56.32±3.08 |
| DDGCL | 65.15±4.54 | 55.21±6.19 | 62.34±5.13 | 50.91±2.08 | 54.96±1.46 | 61.68±0.81 | 55.62±0.32 | 68.49±5.31 | 47.98±1.11 | 55.90±1.13 | 55.18±2.73 | 42.70±3.26 |
| CPDG | 43.56±6.41 | 65.92±6.25 | 50.32±5.06 | 49.89±1.34 | 52.86±0.64 | 59.72±2.53 | 53.82±1.50 | 49.71±2.64 | 47.37±2.23 | 56.40±1.17 | 53.58±2.10 | 40.01±3.59 |
| GRAPHPROMPT | 73.78±5.62 | 60.89±6.37 | 64.60±5.76 | 51.28±2.43 | 55.67±0.26 | 67.46±0.31 | 51.07±0.75 | 86.78±3.14 | 48.46±0.28 | 59.18±0.49 | 50.27±0.58 | 87.45±2.57 |
| PROG | 60.86±7.43 | 68.60±5.64 | 63.18±4.79 | 51.46±2.38 | 92.28±0.21 | 93.32±0.06 | 58.73±1.58 | 86.24±2.87 | 89.75±0.28 | 90.69±0.08 | 56.42±1.95 | 85.43±3.16 |
| TGAT-TIGPROMPT | 69.21±8.88 | 67.70±9.64 | 73.90±6.68 | 51.38±2.72 | 59.54±1.41 | 78.45±1.44 | 51.69±1.24 | 69.71±4.16 | 49.52±0.85 | 65.66±2.68 | 51.58±4.02 | 48.34±3.28 |
| TGN-TIGPROMPT | 44.80±5.45 | 63.75±5.60 | 55.42±3.60 | 50.84±2.75 | 82.04±2.03 | 83.26±2.38 | 65.00±4.73 | 86.25±2.43 | 81.75±1.97 | 79.51±2.58 | 64.98±4.61 | 86.19±3.06 |
| TGAT-DYGPROMPT | **82.09**±6.43 | 73.50±6.47 | **77.78**±5.08 | **52.03**±2.24 | 69.88±0.18 | 90.76±0.09 | 53.92±0.97 | 72.04±4.71 | 52.58±0.23 | 75.20±0.17 | 53.29±0.87 | 50.82±3.67 |
| TGN-DYGPROMPT | 74.47±3.44 | **74.00**±3.10 | 69.06±3.89 | 51.97±2.16 | **94.33**±0.12 | **96.82**±0.06 | **70.17**±0.75 | **87.02**±1.63 | **92.22**±0.19 | **95.69**±0.08 | **69.77**±0.66 | **87.63**±1.97 |

Results are reported in percent. The best method is bolded and the runner-up is underlined.

**Baselines.** The performance of DYGPROMPT is evaluated against state-of-the-art approaches across four main categories. (1) *Conventional DGNNs*: We first consider a discrete-time framework ROLAND (You et al., 2022) that adapts static GNNs to DGNNs by treating the node embeddings at each GNN layer as hierarchical node states and updating them recurrently over time. We employ GCN (Kipf & Welling, 2017) and GAT (Veličković et al., 2018) for ROLAND. Additionally, we compare to continuous-time DGNNs, including TGAT (Rossi et al., 2020), TGN (Xu et al., 2020), TREND (Wen & Fang, 2022), and GraphMixer (Cong et al., 2023). For a fair comparison, these conventional DGNNs are pre-trained on the first 80% of the events as well, and then continually trained on the same training data in each downstream task following our split. (2) *Dynamic graph pre-training*: DDGCL (Tian et al., 2021) and CPDG (Bei et al., 2024) follow the "pre-train, fine-tune" paradigm. They first pre-train a DGNN, leveraging the intrinsic attributes and dynamic patterns of graphs. Then, they fine-tune the pre-trained model for downstream tasks based on task-specific training data. (3) *Static graph prompting*: GraphPrompt (Liu et al., 2023b) and ProG (Sun et al., 2023) employ a static prompt to adapt pre-trained GNNs to downstream tasks. Specifically, we employ DGNNs as the pre-trained backbones—TGAT for GraphPrompt and TGN for ProG—as these combinations perform competitively with their respective prompting approaches. (4) *Dynamic graph prompting*: TIGPrompt (Chen et al., 2024) proposes a prompt generator to generate time-aware prompts. Note that both TIGPrompt and our method DYGPROMPT can be applied to different backbones, such as TGAT and TGN. More details on these baselines are described in Appendix D, with further implementation details for the baselines and DYGPROMPT in Appendix E.

## 5.2 PERFORMANCE COMPARISON WITH BASELINES

We compare the performance of all methods on temporal node classification and link prediction in Table 1 and make the following observations.

First, DYGPROMPT achieves outstanding performance across node classification and link prediction tasks. This observation highlights the effectiveness of our proposed dual prompts and condition-nets. To further understand the impact of specific design choices in DYGPROMPT and its robustness to various backbones, we defer a detailed investigation to the ablation studies in Sect. 5.3 and the backbone analysis in Sect. 5.4. Second, TIGPrompt, the only other dynamic graph prompt learning method, significantly lags behind DYGPROMPT when using the same backbone. While it employs time-aware prompts, it lacks fine-grained node-time characterization and is thus unable to capture complex node-time patterns, where nodes and time mutually influence each other. Third, Graph-Prompt and ProG perform competitively because we employ DGNNs as their backbones in pre-training, rather than static GNNs. However, they still underperform compared to DYGPROMPT, as they only use static prompts.

**REMARK.** It is worth noting that the results of TGAT and TGN in our experiments are lower than those reported in their original papers. A possible reason is that, in their original setting, these methods were trained on data that immediately precede the test set chronologically. In contrast, in our experiments, to maintain consistency with the pre-training setup, we train them in two stages: first with pre-training, and then downstream training on limited task data where we sample only

Table 2: Ablation study reporting AUC-ROC (%), with TGAT as the backbone.

| Methods | Node prompt | Time prompt | NCN | TCN | Node classification Wikipedia | Reddit | MOOC | Transductive Link Prediction Wikipedia | Reddit | MOOC | Inductive Link Prediction Wikipedia | Reddit | MOOC |
|---|---|---|---|---|---|---|---|---|---|---|---|---|---|
| VARIANT 1 | × | × | × | × | 67.00 | 53.64 | 59.27 | 55.78 | 62.43 | 51.49 | 48.21 | 57.30 | 51.42 |
| VARIANT 2 | ✓ | × | × | × | 72.59 | 61.82 | 63.50 | 68.12 | 88.59 | 51.24 | 51.89 | 74.84 | 51.37 |
| VARIANT 3 | × | ✓ | × | × | 73.22 | 62.51 | 62.59 | 66.51 | 87.06 | 51.26 | 50.28 | 69.71 | 50.50 |
| VARIANT 4 | ✓ | ✓ | × | × | 72.25 | 63.11 | 62.87 | 68.36 | 90.31 | 52.17 | 52.56 | 75.13 | 51.33 |
| VARIANT 5 | ✓ | × | ✓ | × | 81.40 | 73.12 | 77.15 | 69.56 | 90.10 | 52.16 | 52.57 | 75.50 | 53.31 |
| VARIANT 6 | × | ✓ | × | ✓ | 80.34 | 72.59 | 76.16 | 66.62 | 87.34 | 51.22 | 49.05 | 73.16 | 52.34 |
| DYGPROMPT | ✓ | ✓ | ✓ | ✓ | 82.09 | 73.50 | 77.78 | 69.88 | 90.76 | 53.92 | 52.58 | 75.20 | 53.29 |

Table 3: AUC-ROC (%) evaluation of DYGPROMPT with different DGNN backbones.

| Pre-training Backbone | Downstream Adaptation | Node classification Wikipedia | Reddit | MOOC | Transductive link prediction Wikipedia | Reddit | MOOC | Inductive link prediction Wikipedia | Reddit | MOOC |
|---|---|---|---|---|---|---|---|---|---|---|
| DYREP | - | 50.61 | 49.54 | 50.33 | 58.45 | 58.02 | 50.29 | 56.87 | 57.25 | 50.34 |
| | DYGPROMPT | **53.52** | **50.98** | **51.62** | **91.64** | **93.84** | **72.04** | **90.28** | **93.08** | **72.27** |
| JODIE | - | 51.37 | 49.80 | 50.53 | 62.40 | **59.81** | 50.71 | 59.59 | **61.28** | 50.57 |
| | DYGPROMPT | **62.84** | **60.93** | **67.84** | **63.56** | 58.89 | **52.06** | **62.80** | 58.17 | **52.33** |
| TGAT | - | 67.00 | 53.64 | 59.27 | 55.78 | 62.43 | 51.49 | 48.21 | 57.30 | 51.42 |
| | DYGPROMPT | **82.09** | **73.50** | **77.78** | **69.88** | **90.76** | **53.92** | **52.58** | **75.20** | **53.29** |
| TGN | - | 50.61 | 49.54 | 50.33 | 72.48 | 67.37 | 54.60 | 74.38 | 69.81 | 54.62 |
| | DYGPROMPT | **74.47** | **74.00** | **69.06** | **94.33** | **96.82** | **70.17** | **92.22** | **95.69** | **69.77** |
| TREND | - | 69.92 | 64.85 | 66.79 | 63.24 | **80.42** | 58.70 | 50.15 | **65.13** | 57.52 |
| | DYGPROMPT | **70.15** | **65.24** | **67.58** | **64.35** | 79.62 | **59.45** | **51.26** | 64.88 | **59.13** |
| GraphMixer | - | 65.43 | 60.21 | 63.72 | 59.73 | 61.88 | 52.42 | 51.34 | **57.64** | 51.16 |
| | DYGPROMPT | **66.39** | **61.42** | **64.18** | **60.25** | **62.31** | **52.94** | **52.19** | 57.43 | **52.55** |

"-" refers to fine-tuning or continually training the backbone on downstream task data without our prompt design.

30 events for each task. Moreover, training data that are temporally closer to the test set are more important than earlier data in pre-training. However, due to limited data in the downstream training stage, the models converge quickly and fail to effectively learn from these crucial recent time intervals, resulting in lower performance than their original setting. In contrast, our prompt-based approach can better handle the temporal gap and adapt well to the test data.

## 5.3 ABLATION STUDIES

To thoroughly understand the impact of each component within DYGPROMPT, we compare DYG-PROMPT with its variants that employ different prompts and condition-nets. These variants and their corresponding results, using TGAT as the backbone, are presented in Table 2 across three datasets.

For node classification, both node and time prompts are beneficial, as Variant 2 (with node prompt) and Variant 3 (with time prompt) outperform Variant 1 (without these prompts). Incorporating both prompts in Variant 4 could further boost the performance. Moreover, the node condition-net is advantageous, as it integrates node characteristics into time features, since Variant 5 outperforms Variant 2. Similarly, the time condition-net is beneficial for capturing temporal subtleties at different times to enhance node feature characterization, as Variant 6 outperforms Variant 3. Lastly, DYGPROMPT achieves the best performance, demonstrating the effectiveness of dual prompts and condition-nets.

For link prediction, the results generally exhibit similar patterns to those in node classification. A notable exception is the MOOC dataset, where the differences between the variants are generally smaller. A potential reason is that interactions in MOOC (online courses) are less dynamic and diverse than interactions in Reddit (a major social network) and Wikipedia (a high-traffic site), reducing the effectiveness of prompts designed to bridge the pre-training and downstream gaps. However, our full model DYGPROMPT still achieves significantly better performance than Variant 1, which does not utilize any prompts. On the other hand, in Reddit and Wikipedia, even though both pre-training and downstream tasks involve link prediction, node prompts remain useful due to the potential distributional differences in the downstream training data across tasks.

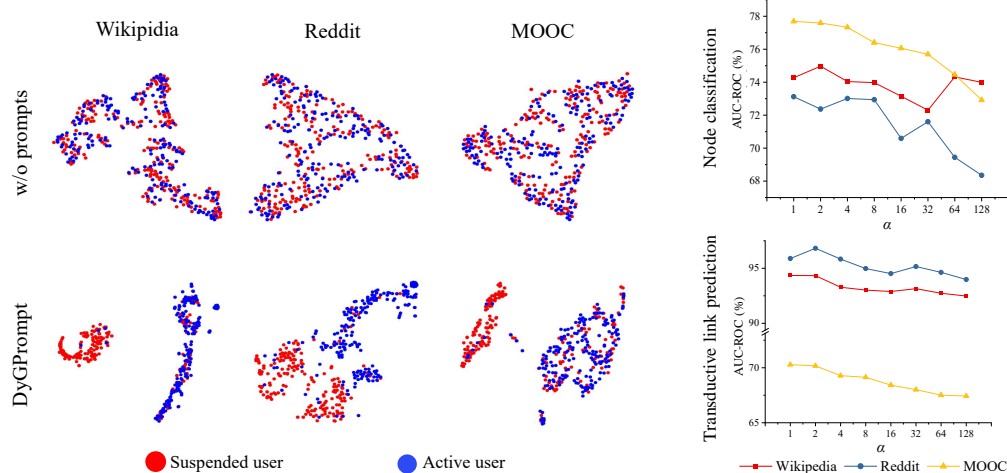

Figure 3: Visualization of output embedding space of nodes.          Figure 4: Sensitivity of $\alpha$.

To further visualize the effect of our prompt design, we sample 200 suspended nodes and 200 active nodes from *Wikipedia*, *Reddit* and *MOOC*, respectively, and show the output embedding space of these nodes, calculated by Variant 1 (w/o prompts) and DYGPROMPT, in Fig. 3. We observe that with both dual prompts and dual condition-nets, node embeddings from different classes are clearly separated, demonstrating the advantages of DYGPROMPT.

### 5.4 IMPACT OF BACKBONES AND HYPERPARAMETERS

First, to analyze the flexibility and robustness of DYGPROMPT, we evaluate its performance on different DGNN backbones, including DYREP (Trivedi et al., 2019), JODIE (Kumar et al., 2019), TGAT, TGN, TREND, and GRAPHMIXER. The results on three datasets are reported in Table 3. We observe that regardless of the backbone utilized, DYGPROMPT surpasses the original backbone without our prompt design in almost all cases, indicating the robustness of our proposed framework.

Next, we investigate the sensitivity of hyperparameters in DYGPROMPT. Particularly, we employ two MLPs with a bottleneck structure as the dual condition-nets (Wen & Fang, 2023; Wu & Lee, 2018). The perceptrons embed the input feature dimension $d$ to $\tilde{d}$, and then restore it back to $d$, where $\tilde{d} = d/\alpha$. Here, $\alpha$ is a hyperparameter that scales the hidden dimension. We evaluate the impact of $\alpha$, as illustrated in Fig. 4. We observe that for both node classification and link prediction tasks, performance generally declines as $\alpha$ increases, likely due to greater information loss from a "smaller bottleneck." The performance when $\alpha \approx 2$ is generally competitive across different datasets. Therefore, in our experiments, we set $\alpha = 2$.

## 6 CONCLUSIONS

In this paper, we explored prompt learning on dynamic graphs, aiming to narrow the gap between pre-training and downstream applications on such graphs. Our proposed approach, DYGPROMPT, employs dual prompts to overcome the divergent objectives and temporal variations across pre-training and downstream tasks. Moreover, we propose dual condition-nets to mutually character-ize node and time patterns. Finally, we conducted extensive experiments on four public datasets, demonstrating that DYGPROMPT significantly outperforms various state-of-the-art baselines.

### ACKNOWLEDGMENTS

This research / project is supported by the Ministry of Education, Singapore, under its Academic Research Fund Tier 2 (Proposal ID: T2EP20122-0041). Any opinions, findings and conclusions or recommendations expressed in this material are those of the author(s) and do not reflect the views of the Ministry of Education, Singapore.

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

---

**Algorithm 1** DOWNSTREAM PROMPT TUNING FOR DYGPROMPT

---

**Input:** Pre-trained DGNN with parameters $\Theta_0$, including those of DGE and TE
**Output:** Optimized dual prompts $(\mathbf{p}^{\text{node}}, \mathbf{p}^{\text{time}})$, and optimized parameters $(\kappa, \phi)$ of dual condition-nets
 1: /* Encoding dynamic graphs and time via pre-trained DGNN */
 2: $\mathbf{p}^{\text{node}}, \mathbf{p}^{\text{time}}, \kappa, \phi \leftarrow$ initialization
 3: **while** not converged **do**
 4:    **for** each dynamic graph $G = (V, E)$ with feature matrix $\mathbf{X}_t$ at time $t$ **do**
 5:        $\mathbf{x}_{t,v} \leftarrow \mathbf{X}_t[v]$, where $v$ is a node in $G$
 6:        /* Dual prompt modification by Eqs. 4 and 5 */
 7:        $\mathbf{x}_{t,v}^{\text{node}} \leftarrow \mathbf{p}^{\text{node}} \odot \mathbf{x}_{t,v}$
 8:        $\mathbf{f}_t^{\text{time}} \leftarrow \mathbf{p}^{\text{time}} \odot \mathbf{f}_t$
 9:        /* Dual conditional prompts generation by Eqs. 6 and 8 */
10:        $\tilde{\mathbf{p}}_t^{\text{node}} \leftarrow \text{TCN}(\mathbf{f}_t^{\text{time}}; \kappa)$
11:        $\tilde{\mathbf{p}}_{t,v}^{\text{time}} \leftarrow \text{NCN}(\mathbf{x}_{t,v}^{\text{node}}; \phi)$
12:        /* Dual conditional prompts modification by Eqs. 7 and 9 */
13:        $\tilde{\mathbf{x}}_{t,v}^{\text{node}} \leftarrow \tilde{\mathbf{p}}_t^{\text{node}} \odot \mathbf{x}_{t,v}^{\text{node}}$
14:        $\tilde{\mathbf{f}}_{t,v}^{\text{time}} \leftarrow \tilde{\mathbf{p}}_{t,v}^{\text{time}} \odot \mathbf{f}_t^{\text{time}}$
15:        /* Temporal node embedding calculation by Eq. 10 */
16:        $\mathbf{h}_{t,v} \leftarrow \text{DGE}\left(\text{Fuse}(\tilde{\mathbf{x}}_{t,v}^{\text{node}}, \tilde{\mathbf{f}}_{t,v}^{\text{time}}), \left\{\text{Fuse}(\tilde{\mathbf{x}}_{t',u}^{\text{node}}, \tilde{\mathbf{f}}_{t`,v}^{\text{time}}) : (u, t') \in \mathcal{N}_v\right\}\right)$
17:        /* Update prototypical nodes */
18:        **for** each class $y$ **do**
19:            $\mathbf{h}_{t,y} \leftarrow \text{AVERAGE}(\mathbf{h}_{t,v}:$ instance $v$ belongs to class $y$ at time $t)$
20:        /* Optimizing dual prompts and the parameters in dual condition-nets */
21:        Calculate $\mathcal{L}_{\text{down}}(\mathbf{p}^{\text{node}}, \mathbf{p}^{\text{time}}, \kappa, \phi)$ by Eq. 11
22:        Update $\mathbf{p}^{\text{node}}, \mathbf{p}^{\text{time}}, \kappa, \phi$ by backpropagating $\mathcal{L}_{\text{down}}(\mathbf{p}^{\text{node}}, \mathbf{p}^{\text{time}}, \kappa, \phi)$
23: **return** $\mathbf{p}^{\text{node}}, \mathbf{p}^{\text{time}}, \kappa, \phi$

---

APPENDICES

## A ALGORITHM

We outline the key steps for downstream prompt tuning in Algorithm 1. In lines 6–8, we use dual prompts to modify node and time features. In lines 9–14, we perform dual conditional prompting. Specifically, we generate node conditioned time prompts and time conditioned node prompts (lines 9–11), and then modify node and time's features using these conditional prompt (lines 12–14). In line 15–16, we calculate node embeddings based on the prompted node and time features. Finally, we update the embeddings for the prototypical nodes based on the few-shot labeled data provided in the task (lines 18–19) and optimize dual prompts and dual condition-nets (lines 20–22). Note that updating prototypical nodes is required exclusively for node classification tasks.

## B COMPLEXITY ANALYSIS

For a downstream dynamic graph $G = (V, E, T)$, the computational process of DYGPROMPT involves two main parts: encoding nodes via a pre-trained TGN and prompt learning. The first part's complexity is determined by the TGN's architecture, similar to other methods using a pre-trained TGN. In a standard TGN, each node aggregates messages from up to $d$ neighbors per layer. Assuming the aggregation involves at most $d$ neighbors, the complexity of calculating node embeddings over $L$ layers per batch time is $O(d^L \cdot |V|)$. The second part, downstream prompt learning, has two stages: dual prompting and dual conditional prompting. For the dual prompting, each node and time feature is adjusted by the node prompt and time prompt, respectively, leading to a complexity of $O(|V| + 1)$ per batch time. For the dual condition-nets with $K$ layers, for each timestamp, we first generate the node-based condition prompt with a complexity of $O(K)$, and the time-based condition prompts with a complexity of $O(K \cdot |V|)$. Then the condition prompts are used to modify nodes and time features, resulting in a complexity of $O(|V| + 1)$. Therefore, the total complexity for prompt learning per batch time is $O((K + 2) \cdot (|V| + 1))$. In conclusion, the overall complexity of DYG-PROMPT is $O(d^L \cdot |V| + (K + 2) \cdot (|V| + 1))$. The first part dominates the overall complexity, as

Table 4: Summary of datasets.

| Dataset | Nodes num | Edges num | Node classes | Dynamic labels | Feature dimension | Time span |
|---------|-----------|-----------|--------------|----------------|-------------------|-----------|
| Wikipedia | 9,227 | 157,474 | 2 | 217 | 172 | 30 days |
| Reddit | 11,000 | 672,447 | 2 | 366 | 172 | 30 days |
| MOOC | 7,144 | 411,749 | 2 | 4,066 | 172 | 30 days |
| Genre | 1,505 | 17,858,395 | 474 | 984 | 86 | 1,500 days |

$O(d^L \cdot |V|)$ far exceeds $O((K+2) \cdot (|V|+1))$. Thus, the additional computational cost introduced by the conditional prompt tuning step is minimal.

## C   FURTHER DESCRIPTIONS OF DATASETS

In this section, we provide a summary of these datasets in Table 4 and a further comprehensive descriptions of these datasets.

(1) *Wikipedia*[2] represents a month of modifications made by contributors on Wikipedia pages (Ferschke et al., 2012). Following prior studies (Rossi et al., 2020; Xu et al., 2020), we utilize data from the most frequently edited pages and active contributors, obtaining a temporal graph containing 9,227 nodes and 157,474 temporal directed edges. Dynamic labels indicate whether contributors temporarily banned from editing.

(2) *Reddit*[3] represents a dynamic network between posts and users on subreddits, where an edge represents a user writes a post to the subreddit, with about 11,000 nodes, about 700,000 temporal edges, and dynamic labels indicating whether user is banned from writing posts.

(3) *MOOC*[4] is a student-course dataset, representing the actions taken by student on MOOC platform. The nodes are users and courses, and edges represent the actions by users on the courses. Dynamic labels indicate whether the student drops-out after the action.

(4) *Genre*[5] is a dynamic network linking users to music genres, with edges represents a user listens to a specific genre at a certain time. The dataset includes 1,505 nodes and 17,858,395 temporal edges, and dynamic labels indicate each user's most favored music genre.

## D   FURTHER DESCRIPTIONS OF BASELINES

In this section, we provide additional details about the baselines used in our experiments.

(1) **Conventional DGNNs**

- **ROLAND** (You et al., 2022): ROLAND adapts static GNNs for dynamic graph learning. It offers a new viewpoint for static GNNs, where the node embeddings at different GNN layers are viewed as hierarchical node states, thus capturing temporal evolution.

- **TGAT** (Rossi et al., 2020): TGAT use a self-attention mechanism and a time encoding technique based on the classical Bochner's theorem from harmonic analysis. By stacking TGAT layers, the network recognizes the node embeddings as functions of time and is able to inductively infer embeddings for both new and observed nodes as the graph evolves.

- **TGN** (Xu et al., 2020): TGN incorporates a memory mechanism that updates node states based on new events, effectively capturing the historical context and dependencies over time. It further propose a novel training strategy that supports efficient parallel processing.

- **TREND** (Wen & Fang, 2022): TREND incorporates the Hawkes process into graph neural networks, and adopts event dynamics and node dynamics to capture the individual charac-

---

[2]http://snap.stanford.edu/jodie/wikipedia.csv
[3]http://snap.stanford.edu/jodie/reddit.csv
[4]http://snap.stanford.edu/jodie/mooc.csv
[5]https://object-arbutus.cloud.computecanada.ca/tgb/tgbn-genre.zip

      teristics of each event and the collective characteristics of events on the same node, respectively.

- **GraphMixer** (Cong et al., 2023): GraphMixer employs a more simple MLP for feature learning, with a portion of the parameters dedicated to time encoding being fixed. This approach enhances the model's ability to capture temporal dynamics while maintaining a simpler and more flexible architecture.

(2) **Graph Pre-training Models**

- **DDGCL** (Tian et al., 2021): DDGCL designs a self-supervised dynamic graph pre-training method via contrasting two nearby temporal views of the same node identity.

- **CPDG** (Bei et al., 2024): CPDG integrates two types of contrastive pre-training strategies to learn comprehensive node representations that encapsulate both long-term and short-term patterns.

(3) **Graph Prompt Models**

- **GraphPrompt** (Liu et al., 2023b): GraphPrompt employs subgraph similarity as a mechanism to unify different pretext and downstream tasks, including link prediction, node classification, and graph classification. A learnable prompt is subsequently tuned for each downstream task.

- **ProG** (Sun et al., 2023): ProG reformulate the node- and edge-level tasks as graph-level tasks, and proposes prompt graphs with specific nodes and structures to guide different task.

- **TIGPrompt** (Chen et al., 2024): TIGPrompt propose a prompt generator that generates personalized, time-aware prompts for each node, enhancing the adaptability and expressiveness of node embeddings for downstream tasks.

## E    IMPLEMENTATION DETAILS

**Environment.** The environment in which we run experiments is:

- Operating system: Windows 11
- CPU information: 13th Gen Intel(R) Core(TM) i5-13600KF
- GPU information: GeForce RTX 4070Ti (12 GB)

**Optimizer.** For all experiments, we use the Adam optimizer.

**Details of baselines.** For all open-source baselines, we utilize the officially provided code. For the non-open-source CPDG and TIGPrompt, we use our own implementations. Each model is tuned according to the settings recommended in their respective literature to achieve optimal performance.

For both GCN and GAT in Roland, we employ a 2-layer architecture.

For TGAT and TGN, we sample 20 temporal neighbors per node to update their representations. For Trend, after sampling neighboring nodes, we apply Hawkes processing to these temporal neighbors using different time decay factors based on the time of each event. For GraphMixer, we employ an MLP to process the input nodes along with their positive and negative examples. The output is then passed through a network composed entirely of linear layers for prediction during training.

For DDGCL, we compares two temporally adjacent views of each node, utilizing a time-dependent similarity evaluation and a GAN-style contrastive loss function. For CPDG, we employ depth-first-search and breadth-first-search to sample neighbors of each nodes.

For GraphPrompt, we employ 1-hop subgraph for similarity calculation. For TIGPrompt, we employ projection prompt generator, as it is reported to yield the best performance in their literature.

For all baselines, we set the hidden dimension to 172 for *Wikipedia*, *Reddit*, and *MOOC*, and to 86 for *Genre*. DDGCL, CPDG, and GraphPrompt leverage TGAT as their backbone, while ProG and

Table 5: AUC-ROC (%) evaluation for anomaly detection on a large-scale dataset DGraph.

| Method | DGraph |
|---|---|
| GRAPHPROMPT | $60.18 \pm 13.06$ |
| TIGPROMPT | $64.42 \pm 10.37$ |
| DYGPROMPT | $\mathbf{73.39} \pm 5.85$ |

Table 6: Prompt-tuning and testing runtimes (seconds) on node classification.

| Method | Wikipedia | | Reddit | | DGraph | |
|---|---|---|---|---|---|---|
| | Training | Testing | Training | Testing | Training | Testing |
| GRAPHPROMPT | 0.147 | 0.230 | 0.128 | 0.251 | 0.436 | 0.324 |
| TIGPROMPT | 0.312 | 0.397 | 0.356 | 0.411 | 0.847 | 1.063 |
| DYGPROMPT | 0.273 | 0.233 | 0.134 | 0.274 | 0.574 | 0.633 |

TIGPrompt use TGN. For TIGPrompt, we evaluate its performance using both TGAT and TGN as the backbone.

**Details of DYGPROMPT.** For our proposed DYGPROMPT, We conducted experiments using TGN and TGAT as backbones. We employ a dual-layer perceptrons with bottleneck structure as the condition-net, and set the hidden dimension of the condition net as 86 for *Wikipedia*, *Reddit* and *MOOC*, while 43 for *Genre*. We set the hidden dimension to 172 for *Wikipedia*, *Reddit*, and *MOOC*, and to 86 for *Genre*.

# F  ADDITIONAL EXPERIMENTS

## F.1  PERFORMANCE ON LARGE-SCALE DATASET

We evaluate DYGPROMPT under the same setting introduced in Sect. 5.1 on a large-scale dataset DGraph (Huang et al., 2022), which consists of 3,700,550 nodes, 4,300,999 edges, and 1,225,601 labeled nodes. The dataset is designed for the anomaly detection task, a form of binary node classification. The results are shown in Table 5. We observe that DYGPROMPT outperforms competitive baselines, showing the effectiveness of DYGPROMPT on large-scale datasets.

## F.2  EFFICIENCY ANALYSIS

We compare the prompt tuning and testing runtimes between DYGPROMPT and competitive baselines for the node classification task. As shown in Table 6, DYGPROMPT takes only marginally longer than GraphPrompt, while requiring less time than TIGPrompt. The slight overhead over GraphPrompt is acceptable given the substantial improvement in performance. Note that the testing runtimes are measured on the entire test set, which can be much larger than the samples used for prompt tuning, resulting in generally longer runtimes compared to prompt tuning.

## F.3  PERFORMANCE USING THE SPLITS OF TIGPROMPT

We adopt the splits of TIGPrompt and conduct further experiments, using 50% of the data for pre-training and 20% for prompt tuning, with the remaining 30% equally divided for validation and testing. We report the results in Table 7, where DYGPROMPT still consistently outperforms TIG-Prompt.

## F.4  PERFORMANCE ON FIVE-WAY CLASSIFICATION

While most of our datasets involve binary classification, we also conduct five-way node classification on the ML-Rating dataset (Harper & Konstan, 2015) with 9,995 nodes, 1,000,210 edges, and 5 classes. We adopt the same setting introduced in Sect. 5.1 and compare DYGPROMPT with two

Table 7: AUC-ROC (%) evaluation using the splits of TIGPrompt (Chen et al., 2024).

| Method | Node classification | | Transductive link prediction | | Inductive link prediction | |
|---|---|---|---|---|---|---|
| | Wikipedia | MOOC | Wikipedia | MOOC | Wikipedia | MOOC |
| TIGPROMPT | 78.85±1.35 | 63.40±2.31 | 93.95±0.47 | 78.98±0.52 | 91.35±0.38 | 80.26±0.76 |
| DYGPROMPT | **81.31**±1.13 | **64.58**±1.95 | **97.78**±0.36 | **87.15**±0.42 | **96.73**±0.42 | **86.14**±0.39 |

Table 8: AUC-ROC (%) evaluation for five-way node classification on ML-Rating.

| Method | ML-Rating |
|---|---|
| GRAPHPROMPT | $52.37 \pm 1.29$ |
| TIGPROMPT | $53.26 \pm 1.33$ |
| DYGPROMPT | $\mathbf{54.82} \pm 1.27$ |

competitive baselines in Table 8. DYGPROMPT consistently outperforms these baselines, demonstrating its effectiveness.

## G  FURTHER COMPARISON WITH RELATED WORK

We summarize the comparison of DYGPROMPT with representative graph prompt learning methods, including GraphPrompt (Liu et al., 2023b) and ProG (Sun et al., 2023) for static graphs, and the contemporary work TIGPrompt (Chen et al., 2024) for dynamic graphs (using its best-performing variant, Projection TProG, for illustration), as shown in Table 9.

More specifically, first, a node prompt modifies the node features to reformulate the task input and bridge the task gap (Eq. 4), while a time prompt adjusts the time features to capture the temporal evolution of the dynamic graph (Eq. 5). The dual prompts are motivated by Challenge 1 in Sect. 1. Previous static graph prompt learning methods (GraphPrompt and ProG) only utilize a node prompt, neglecting the temporal gap between pre-training and downstream tasks. The dual prompts also demonstrate empirical benefits. As shown in the ablation study in Table 2, Variant 2 (using only node prompt) and Variant 3 (using only time prompt) consistently outperform Variant 1. Additionally, Variant 4 (incorporating both node and time prompts) generally outperforms Variants 2 and 3, demonstrating the effectiveness of node and time prompts in bridging the task gap and temporal gap between pre-training and downstream tasks.

Second, time-aware node prompts adjust node prompts, and ultimately node features, to reflect temporal influence on the node prompts (Eq. 7) instead of using a fixed node prompt across time. On the other hand, node-aware time prompts adjust the time prompt for each time feature by incorporating node-specific characteristics (Eq. 9) instead of using a fixed time prompt across nodes. Consequently, node-time patterns and their dynamic interplay are captured, addressing the Challenge 2 in Section 1. Static graph prompt learning methods lack such designs, failing to capture the mutual characterization between node and time patterns. Moreover, TIGPrompt considers only the temporal impact on nodes by generating time-aware node prompts, but neglects that time prompts for each node can also be influenced by node features, hindering its ability to model the scenario where different nodes may exhibit divergent behaviors even at the same time point, as illustrated in Fig. 1(a). As shown in the ablation study in Table 2, Variant 5 (with node-aware time prompts) outperforms Variant 2 (without node-aware time prompts), Variant 6 (with time-aware node prompts) outperforms Variant 3 (without time-aware node prompts), and DyGPrompt (with both) outperforms Variant 4 (without either), further demonstrating the effectiveness of our designs.

Third, we propose dual condition-nets to effectively generate time-aware node prompts (Eq. 6) and node-aware time prompts (Eq. 8) while avoid overfitting in a parameter-efficient way. These condition-nets generate prompts conditioned on the input features rather than directly parameterizing the prompts, significantly reducing the number of learnable parameters in the downstream prompt-tuning phase. In our implementation, we use a 2-layer MLP as the condition-net. Specifically, our dual condition-nets contains merely 3,104 learnable parameters for Wikipedia, Reddit and MOOC, 1,554 for Genre and 8,100 for DGraph, to generate all the time-aware node prompts and node-aware time prompts. In contrast, to generate time-aware node prompts, TIGPrompt requires

Table 9: Comparison with representative prompt learning methods.

| | Explicit node prompt | Explicit time prompt | Time-aware node prompts | Node-aware time prompts | Condition-net |
|---|---|---|---|---|---|
| GRAPHPROMPT | ✓ | ✗ | ✗ | ✗ | ✗ |
| PROG | ✓ | ✗ | ✗ | ✗ | ✗ |
| TIGPROMPT | ✗ | ✗ | ✓ | ✗ | ✗ |
| DYGPROMPT | ✓ | ✓ | ✓ | ✓ | ✓ |

learning a specific prompt for each node, leading to a significantly larger number of learnable parameters: 1,609,274 for Wikipedia, 1,911,478 for Reddit, 1,250,998 for MOOC, 130,826 for Genre, and 66,610,156 for DGraph. Moreover, as shown in Table 6, DYGPROMPT requires less training and testing time compared to TIGPrompt, further demonstrating the efficiency of using condition-net. Thus, the condition-nets in DYGPROMPT is another significant difference from previous work, addressing Challenge 2 in a parameter-efficient manner.

