# OpenReview forum: "Node-Time Conditional Prompt Learning in Dynamic Graphs"
_ICLR.cc/2025/Conference — ICLR 2025 Poster_

### Official Review · Reviewer_obkg · 2024-10-21

**Soundness:** 3
**Presentation:** 3
**Contribution:** 3
**Rating:** 6
**Confidence:** 4

**Summary:**

The paper introduces DYGPROMPT, a novel framework for dynamic graph modeling that aims to bridge the gap between pre-training and downstream tasks, such as node classification and link prediction, on dynamic graphs. The authors propose a two-stage approach involving pre-training and prompt learning. In the pre-training phase, a dynamic graph neural network is trained on the task of temporal link prediction. For the downstream tasks, the authors design dual prompts (node and time prompts) to address the differences in objectives and temporal variations between the pre-training and downstream phases. Additionally, they introduce dual condition nets to generate a series of time-conditioned node prompts and node-conditioned time prompts, capturing the evolving node-time patterns. The framework is evaluated on four public datasets, demonstrating superior performance compared to state-of-the-art methods.

**Strengths:**

S1. The paper presents a creative solution to the challenge of adapting pre-trained DGNNs to downstream tasks, which is a novel contribution to dynamic graph learning.

S2. The paper is well-written and easy to follow.

S3. Extensive experiments compared with DGNNs, SOTA pre-training models, and graph prompt learning models demonstrate the effectiveness of the proposed model.

**Weaknesses:**

W1.  While the paper mentions the computational complexity, a deeper analysis comparing the complexity of DYGPROMPT with other methods, especially in terms of training and inference time, is necessary for a prompt-based model.

W2. The used datasets are very small-scale. The largest dataset used only contains about 10K nodes. This makes it difficult to confirm whether the model can be successfully applied to large-scale data and the real world.

W3. The experimental datasets have extreme class numbers. Specifically, 3 of the datasets are binary classification and another one contains 474 classes. It would be better to experiment on other more datasets.

**Questions:**

Q1: Can the authors comment on the scalability of DYGPROMPT to large-scale dynamic graphs, both in terms of computational resources and performance?

Q2: How does DYGPROMPT adapt to long-term temporal changes in dynamic graphs, especially when the graph's structure and node behaviors evolve significantly over time?

---

> ### Author Response · Authors · 2024-11-21
>
> We greatly appreciate your acknowledgment of our paper and your insightful review, which has helped us improve our work. We address your concerns as follows and we are happy to discuss any further questions you may have.
>
> >**W1**: Training and inference time
>
> Following your advice, we further compared the prompt tuning and testing times between DyGPrompt and competitive baselines for node classification tasks. We also included a large-scale dataset, DGraph [1] with 3,700,550 nodes and 4,300,999  edges (see our response to **W2 & W3 & Q1**), for a more comprehensive comparison. As shown below (in seconds), DyGPrompt takes only marginally longer than GraphPrompt, while requiring less time than TIGPrompt. The slight overhead over GraphPrompt is acceptable given the substantial improvement in performance. (Also note that the testing times are measured on the entire test set, which can be larger than the samples used for prompt tuning, resulting in generally longer testing times compared to prompt tuning.)
>
> We have also updated the paper and **included these results in Appendix A.7**.
>
> |  | Wikipedia training | Wikipedia testing | Reddit training | Reddit testing | DGraph training | DGraph testing|
> | ------ | -------------- | ------------------- | ----------- | ---------------- |-|-|
> | GraphPrompt  | 0.147| 0.230|0.128| 0.251|0.436 | 0.324|
> | TIGPrompt    |0.312  |0.397 |0.356 |0.411|0.847 | 1.063|
> | DyGPrompt    |0.273 |0.233 |0.134 |0.274 |0.574 | 0.633|
>
>
> >**W2 & W3 & Q1**: New large-scale dataset
>
> Following your advice, we evaluate DyGPrompt under the same setting as introduced in Section 5.1 on a new large-scale dataset DGraph [1] (3,700,550 nodes, 4,300,999 edges), designed for dynamic graph anomaly detection (a form of node classification). We report the AUC-ROC (%) in the table below, and DyGPrompt still outperforms competitive baselines, showing the robustness of DyPrompt on large-scale dataset. We also updated the paper and include these additional results into Appendix A.6. The training and testing time evaluation on DGraph is shown in our response to **W1**.
>
>
> | | AUC-ROC (%) |
> |-|-|
> |GraphPrompt|60.18±13.06|
> |TIGPrompt|64.42±10.37|
> |DyGPrompt|**73.39**±5.85|
>
>
> >**Q2**: Long-term temporal change
>
> In long-term temporal changes, the node-time pattern may vary significantly.
> In our work, we propose dual condition-nets to generate a set of node and time prompts, capturing the evolving node-time patterns. These time-conditioned node prompts can more flexibly adjust node features to reflect temporal variations, and node-conditioned time prompts incorporate node-specific characteristics to better tailor to distinct node behaviors that may occur at the same time. Consequently, DyGPrompt can effectively adapt node-time patterns to downstream tasks, addressing the limitation of previous works. Specifically, on the long-term dataset Genre (over a span of 1500 days), DyGPrompt outperforms all other baselines, demonstrating that it is able to capturing long-term patterns.
>
> [1] Huang et al. "Dgraph: A large-scale financial dataset for graph anomaly detection." NeurIPS 2022.

---

> ### Author Response · Authors · 2024-11-25
>
> We thank you once again for recognizing our contribution and for raising insightful points that have helped us improve our paper. As the discussion period nears its end, we would like to confirm if our response has adequately addressed your concerns. Based on your comments, we have:
>
> - Compared the training and testing times with competitive baselines.
> - Conducted experiments on a large-scale dataset.
> - Discussed the ability of DyGPrompt to capture long-term temporal changes.
>
> We truly value your insightful feedback and suggestions, which have greatly contributed to improving our work. We look forward to your response.

---

> > ### Comment · Reviewer_obkg · 2024-11-27
> >
> > Thanks for your reply. Some of my concerns have been addressed.
> >
> > However, for W3, I think that Dgraph is also a binary classification dataset on graph anomaly detection tasks (like 0: anomaly, 1: normal).

---

> > > ### Author Response · Authors · 2024-11-28
> > >
> > > Thank you for your advice. We conducted an additional experiment on a node classification task under the same setting as introduced in Section 5.1 on the ML-Rating dataset [2] ( 9,995 nodes, 1,000,210 edges, and 5 classes). We compared DyGPrompt with two competitive baselines, and the results are shown below. DyGPrompt consistently outperforms these baselines, demonstrating its effectiveness. We also updated the paper and include these additional results into Appendix A.10. We hope this has addressed the concern in W3.
> > >
> > > |Method|AUC-ROC (%)|
> > > |-|-|
> > > |GraphPrompt|52.37+1.29|
> > > |TIGPrompt|53.26+1.33|
> > > |DyGPrompt|**54.82**+1.27|
> > >
> > > [2] Harper F M et al. The movielens datasets: History and context. Acm transactions on interactive intelligent systems 2015.

---

> ### Author Response · Authors · 2024-12-02
>
> We truly appreciate your time in reviewing our paper once again. As the discussion period comes to an end, we would like to confirm whether our responses  have effectively addressed your concerns. If you have any remaining questions, we are more than willing to provide further explanations.

---

> > ### Comment · Reviewer_obkg · 2024-12-03
> >
> > Thanks for your reply. I decide to maintain my positive score as the final rating.

---

### Official Review · Reviewer_ngqA · 2024-11-03

**Soundness:** 3
**Presentation:** 4
**Contribution:** 3
**Rating:** 6
**Confidence:** 4

**Summary:**

To address the challenges of prompt-based learning on dynamic graphs, this paper develops a novel pretraining and prompt learning framework for dynamic graph modeling. The authors propose dual prompts and dual condition networks to model the evolving node-time patterns. Extensive experiments are conducted to evaluate the performance of the proposed node-time solution. Overall, the approach is sound and reasonable.

**Strengths:**

1.	This paper identifies the issue of pretrained Graph Neural Networks (GNNs)—specifically, the gap between pre-training and task objectives—and proposes novel prompt learning approaches for dynamic graphs. It also considers the evolving interplay patterns between nodes and time points.
2.	The framework addresses temporal variations across time and divergent task objectives by leveraging a node prompt and a time prompt to reduce the gap between the pre-training and downstream phases. The main novelty lies in the introduction of the time prompt, which narrows the inconsistencies that arise from varying priorities at different times. The proposed method is reasonable.
3.	Extensive experiments are conducted to evaluate the performance of various methods. The source codes are available anonymously, which adds to the credibility of the results.
4.	The manuscript is well written and easy to follow.

**Weaknesses:**

1.	The concept of temporal prompts for dynamic graphs has been previously investigated (e.g., TIGPrompt), but the differences between approaches are not clearly articulated. In the related work, it is stated that “it only considers the temporal factor in node features, overlooking that temporal patterns are also influences by node features.” More explanations are needed to highlight the technical novelty of this paper.
2.	The overall framework presented in Figure 2 could be improved; it is not clear how to compute the downstream loss (Eq. 11) and why it only links to the time branch.
3.	All evaluated datasets are relatively small, and the paper lacks an efficiency or complexity analysis for larger-scale datasets, which is important for practical applications.
4.	There are minor typos and errors, such as “are also influences by node.”

**Questions:**

The study considers only two tasks on dynamic graphs. Is it possible to apply this framework to other graph tasks?

**Details Of Ethics Concerns:**

N.A.

---

> ### Author Response · Authors · 2024-11-21
>
> We sincerely thank you for acknowledging our work and providing positive feedback. We address your concerns as follows and we are glad to discuss any further questions you may have.
>
> > **W1**: Difference with related work
>
> We summarize the comparison of DyGPrompt with representative graph prompt learning methods, including GraphPrompt [1] and ProG [2] for static graphs, and the contemporary work TIGPrompt [3] for dynamic graphs (using its best-performing variant, Projection TProG, for illustration), as below.
>
> | |Explicit node prompt|Explicit time prompt|Time-aware node prompts|Node-aware time prompts|Condition-net|
> |-|-|-|-|-|-|
> |GraphPrompt|$\checkmark$|$\times$|$\times$|$\times$|$\times$|
> |ProG|$\checkmark$|$\times$|$\times$|$\times$|$\times$|
> |TIGPrompt|$\times$|$\times$ |$\checkmark$|$\times$ |$\times$|
> |DyGPrompt| $\checkmark$|$\checkmark$ | $\checkmark$| $\checkmark$|$\checkmark$|
>
> More specifically, first, a *node prompt* modifies the node features to reformulate the task input and bridge the task gap (Eq. 4), while a *time prompt* adjusts the time features to capture the temporal evolution of the dynamic graph (Eq. 5). The dual prompts are motivated by **Challenge 1 in Section 1**. Previous static graph prompt learning methods (GraphPrompt and ProG) only utilize a node prompt, neglecting the temporal gap between pre-training and downstream tasks. Furthermore, TIGPrompt, falls short of explicitly addressing each gap individually.
> The dual prompts also demonstrate empirical benefits. As shown in the ablation study in Table 2, Variant 2 (using only node prompt) and Variant 3 (using only time prompt) consistently outperform Variant 1. Additionally, Variant 4 (incorporating both node and time prompts) generally outperforms Variants 2 and 3, demonstrating the effectiveness of node and time prompts in bridging the task gap and temporal gap between pre-training and downstream tasks.
>
> Second, *time-aware node prompts* adjust node prompts, and ultimately node features, to reflect temporal influence on the node prompts (Eq. 7) instead of using a fixed node prompt across time. On the other hand, *node-aware time prompts* adjust the time prompt for each node by incorporating node-specific characteristics (Eq. 9) instead of using a fixed time prompt across nodes. Consequently, node-time patterns and their dynamic interplay are captured, addressing **Challenge 2 in Section 1**. Static graph prompt learning methods lack such designs, failing to capture the mutual characterization between node and time patterns. Moreover, TIGPrompt considers only the temporal impact on nodes by generating time-aware node prompts, but neglects that time prompts for each node can also be influenced by node features, hindering its ability to model the scenario where different nodes may exhibit divergent behaviors even at the same time point (as illustrated in Fig.1(a)).
> As shown in the ablation study in Table 2, Variant 5 (with time-aware node prompts) outperforms Variant 2 (without time-aware node prompts), Variant 6 (with node-aware time prompts) outperforms Variant 3 (without node-aware time prompts), and DyGPrompt (with both) outperforms Variant 4 (without either), further demonstrating the effectiveness of our designs.
>
> Third, we propose *dual condition-nets* to effectively generate time-aware node prompts (Eq. 6) and node-aware time prompts (Eq. 8) while avoid overfitting in a parameter-efficient way. These condition-nets generate prompts conditioned on the input features rather than directly parameterizing the prompts, significantly reducing the number of learnable parameters in the downstream prompt-tuning phase.
> In our implementation, we use a 2-layer MLP as the condition-net. Specifically, our dual condition-nets contains merely 3,104 learnable parameters for Wikipedia, Reddit and MOOC, 1,554 for Genre and 8,100
> for DGraph, to generate all the time-aware node prompts and node-aware time prompts.
> In contrast, to generate time-aware node prompts, TIGPrompt requires learning a specific prompt for each node, leading to a significantly larger number of learnable parameters: 1,609,274 for Wikipedia, 1,911,478 for Reddit, 1,250,998 for MOOC, 130,826 for Genre, and 66,610,156 for DGraph. Moreover, as shown in our response to **W3**, DyGPrompt requires less training and testing time compared to TIGPrompt, further demonstrating the efficiency of using condition-net. Thus, the condition-nets in DyGPrompt is **another significant difference from previous work**, addressing **Challenge 2 in a parameter-efficient manner**.
>
> Finally, we have updated the paper by **revising parts of the introduction** and including a detailed comparison with related work in **Appendix A.9**.

---

> ### Author Response · Authors · 2024-11-21
>
> >**W2**: Fig.2
>
> Following your advice, we have further improved Fig. 2 by including the downstream loss and highlighting that this loss is computed based on the output of the DGNN, which takes both time and node features as input.
>
> >**W3**: Large-scale dataset and efficiency
>
> **- Performance on large-scale dataset**
>
> Following your advice, we evaluate DyGPrompt under the same setting as introduced in Section 5.1 on a large-scale dataset DGraph [4] (3,700,550 nodes, 4,300,999 edges), designed for dynamic graph anomaly detection (node classification). We report the AUC-ROC (%) in the table below, and DyGPrompt still outperforms competitive baselines, showing the robustness of DyPrompt on large-scale dataset. We also updated the paper and **include these additional results into Appendix A.6**.
>
> | | AUC-ROC (%) |
> |-|-|
> |GraphPrompt|60.18±13.06|
> |TIGPrompt|64.42±10.37|
> |DyGPrompt|**73.39**±5.85|
>
> **- Efficiency of DyGPrompt**
> While a complexity analysis has been provided in Appendix A.2, we further compared the prompt tuning and testing times between DyGPrompt and competitive baselines for node classification tasks. As shown below (in seconds), DyGPrompt takes only marginally longer than GraphPrompt, while requiring less time than TIGPrompt. The slight overhead over GraphPrompt is acceptable given the substantial improvement in performance. (Also note that the testing times are measured on the entire test set, which can be larger than the samples used for prompt tuning, resulting in generally longer testing times compared to prompt tuning.)
>
> We have also updated the paper and **included these results in Appendix A.7**.
>
> | | Wikipedia training | Wikipedia testing | Reddit training | Reddit testing | DGraph training | DGraph testing|
> | ------ | -------------- | ------------------- | ----------- | ---------------- |-|-|
> | GraphPrompt  | 0.147| 0.230|0.128| 0.251|0.436 | 0.324|
> | TIGPrompt    |0.312  |0.397 |0.356 |0.411|0.847 | 1.063|
> | DyGPrompt    |0.273 |0.233 |0.134 |0.274 |0.574 | 0.633|
>
>
>
> >**W4**: Typo errors
>
> Thank you for pointing this out! We have revised the typo errors.
>
> >**Q1**: Other tasks
>
> Node classification and link prediction are the most common tasks in dynamic graph learning. Our method can also be easily extended to graph classification tasks by performing a readout operation on graphs after obtaining the node embeddings. However, as stated in Footnote 2, *"As dynamic graphs focus on the evolving structures within a graph, graph classification is rarely evaluated as a task"*[3,4,5,6,7,8].
>
>
> [1] Liu et al. "Graphprompt: Unifying pre-training and downstream tasks for graph neural networks." WWW 2023.\
> [2] Sun et al. "All in one: Multi-task prompting for graph neural networks." SIGKDD 2023.\
> [3] Chen et al. "Prompt learning on temporal interaction graphs". arXiv 2024.\
> [4] Huang et al. "Dgraph: A large-scale financial dataset for graph anomaly detection." NeurIPS 2022.\
> [5] Xu et al. "Inductive Representation Learning on Temporal Graphs". ICLR 2020.\
> [6] Rossi et al. "Temporal graph networks for deep learning on dynamic graphs". arXiv 2020.\
> [7] Skarding et al. "Foundations and modeling of dynamic networks using dynamic graph neural networks: A survey." iEEE Access 2021.\
> [8] Pareja et al. "Evolvegcn: Evolving graph convolutional networks for dynamic graphs." AAAI 2020.

---

> ### Author Response · Authors · 2024-11-25
>
> Thank you again for acknowledging our contribution and raising insightful points to help improve our paper. As the discussion period is coming to an end, we would like to check whether our response has addressed your concerns. Following your comments, we have:
>
> - Provided a detailed comparison with related work.
> - Improved Fig. 2 for better clarity and understanding.
> - Conducted experiments on a large-scale dataset and compared the training and testing times with competitive baselines.
> - Polished the paper to enhance readability.
> - Discussed the applicability to other tasks.
>
> We genuinely appreciate your valuable feedback and suggestions, which have significantly helped enhance our paper. We look forward to hearing back from you.

---

> > ### Comment · Reviewer_ngqA · 2024-11-29
> >
> > Thanks for the responses. I have reviewed these responses and the comments from other reviewers. I have no further questions and would like to maintain my current positive rating.

---

### Official Review · Reviewer_BsEP · 2024-11-04

**Soundness:** 2
**Presentation:** 2
**Contribution:** 2
**Rating:** 6
**Confidence:** 4

**Summary:**

The paper introduces DYGPROMPT, a novel pretraining and prompt learning framework designed for dynamic graph learning. It addresses the challenges of bridging the gap between pre-training objectives and downstream tasks. The authors propose dual prompts -- node prompts and time prompts -- to address the discrepancies in both task objectives and temporal variations between pre-training and downstream tasks. The authors introduce dual condition-nets to model the evolving node-time patterns in downstream tasks. This includes a time condition-net that generates time-conditioned node prompts and a node condition-net that generates node-conditioned time prompts. Experiments on four public datasets demonstrate its superior performance compared to state-of-the-art approaches in tasks such as temporal node classification and temporal link prediction.

**Strengths:**

- **Relevance to Real-World Applications:** The framework's focus on dynamic graphs makes it highly relevant to real-world applications such as social networks, online learning platforms, and crowdsourcing projects.

- **Parameter Efficiency:** DYGPROMPT's ability to perform well with minimal parameter updates is a significant strength, especially for applications where labeled data is scarce.

- **Open source:** It is commendable that the authors have made their code publicly available, enhancing the transparency and reproducibility of their work.

**Weaknesses:**

- **Motivation:** The motivation for this work is insufficiently justified. After reading the paper, it remains unclear why temporal patterns would be influenced by node features. Time, as a continuous scale, is inherently objective and independent of specific events.

- **Novelty:** Based on the above, the novelty of this work appears limited. TIGPrompt has already proposed an effective strategy for dynamic graph learning by fusing prompts through the concatenation of node and time embeddings for prompt-based fine-tuning. This contrasts with the authors’ claim that TIGPrompt only considers temporal factors within node features. Thus, the main difference between this work and TIGPrompt seems to lie in the adoption of a more complex information fusion strategy rather than a simple concatenation.

- **Experiments:** The experimental setup and results raise some concerns. First, the reported results are noticeably lower than those of TIGPrompt. Given the relevance of TIGPrompt as a baseline, I recommend that the authors adopt identical experimental settings to ensure comparability. Additionally, the influence of model size on performance is unclear; I suggest that the authors report the number of parameters for both their model and TIGPrompt.

- **Writing:** The writing could be further polished to improve clarity and readability.

**Questions:**

See above.

---

> ### Author Response · Authors · 2024-11-21
>
> We thank you for your detailed and insightful feedback. We address your concerns as follows and are happy to provide further clarification if any questions or concerns remain after our response.
>
> > **W1**: Motivation
>
> We believe there is a misunderstanding. While time itself is inherently objective and independent of specific events, **our point here is that the time prompt for node $v$ at time $t$ should be characterized by the specific node $v$, instead of using a fixed time prompt for all nodes.**
> When we say "time features can be influenced by node features", we mean that the time prompt at a specific time can be tailored to reflect the characteristics of each node. As shown in Fig. 1, whether a user engages in a certain topic at a given time depends on the type of user, motivating the need for user-specific time prompts, as called for in Challenge 2 in Section 1.
>
> More specifically, we generate user-specific time prompts using the node condition net in Eq. 8. Hence, even at the same time $t$, the time prompts  become specific to each user $v$ to better tailor to node-specific patterns, while the time encoder TE remains objective and is not influenced by users or events.
>
> To avoid misunderstandings and improve clarity, we have updated the relevant parts of the paper, including parts of Section 1 and Section 4.4.

---

> ### Author Response · Authors · 2024-11-21
>
> > **W2**: Novelty
>
> We summarize the comparison of DyGPrompt with representative graph prompt learning methods, including GraphPrompt [1] and ProG [2] for static graphs, and the contemporary work TIGPrompt [3] for dynamic graphs (using its best-performing variant, Projection TProG, for illustration), as below.
>
> | |Explicit node prompt|Explicit time prompt|Time-aware node prompts|Node-aware time prompts|Condition-net|
> |-|-|-|-|-|-|
> |GraphPrompt|$\checkmark$|$\times$|$\times$|$\times$|$\times$|
> |ProG|$\checkmark$|$\times$|$\times$|$\times$|$\times$|
> |TIGPrompt|$\times$|$\times$ |$\checkmark$|$\times$ |$\times$|
> |DyGPrompt| $\checkmark$|$\checkmark$ | $\checkmark$| $\checkmark$|$\checkmark$|
>
> More specifically, first, a *node prompt* modifies the node features to reformulate the task input and bridge the task gap (Eq. 4), while a *time prompt* adjusts the time features to capture the temporal evolution of the dynamic graph (Eq. 5). The dual prompts are motivated by **Challenge 1 in Section 1**. Previous static graph prompt learning methods (GraphPrompt and ProG) only utilize a node prompt, neglecting the temporal gap between pre-training and downstream tasks. Furthermore, TIGPrompt, falls short of explicitly addressing each gap individually.
> The dual prompts also demonstrate empirical benefits. As shown in the ablation study in Table 2, Variant 2 (using only node prompt) and Variant 3 (using only time prompt) consistently outperform Variant 1. Additionally, Variant 4 (incorporating both node and time prompts) generally outperforms Variants 2 and 3, demonstrating the effectiveness of node and time prompts in bridging the task gap and temporal gap between pre-training and downstream tasks.
>
> Second, *time-aware node prompts* adjust node prompts, and ultimately node features, to reflect temporal influence on the node prompts (Eq. 7) instead of using a fixed node prompt across time. On the other hand, *node-aware time prompts* adjust the time prompt for each node by incorporating node-specific characteristics (Eq. 9) instead of using a fixed time prompt across nodes. Consequently, node-time patterns and their dynamic interplay are captured, addressing **Challenge 2 in Section 1**. Static graph prompt learning methods lack such designs, failing to capture the mutual characterization between node and time patterns. Moreover, TIGPrompt considers only the temporal impact on nodes by generating time-aware node prompts, but neglects that time prompts for each node can also be influenced by node features, hindering its ability to model the scenario where different nodes may exhibit divergent behaviors even at the same time point (as illustrated in Fig. 1).
> As shown in the ablation study in Table 2, Variant 5 (with time-aware node prompts) outperforms Variant 2 (without time-aware node prompts), Variant 6 (with node-aware time prompts) outperforms Variant 3 (without node-aware time prompts), and DyGPrompt (with both) outperforms Variant 4 (without either), further demonstrating the effectiveness of our designs.
>
> Third, we propose *dual condition-nets* to effectively generate time-aware node prompts (Eq. 6) and node-aware time prompts (Eq. 8) while avoid overfitting in a parameter-efficient way. These condition-nets generate prompts conditioned on the input features rather than directly parameterizing the prompts, significantly reducing the number of learnable parameters in the downstream prompt-tuning phase.
> In our implementation, we use a 2-layer MLP as the condition-net. Specifically, our dual condition-nets contains merely 3,104 learnable parameters for Wikipedia, Reddit and MOOC, 1,554 for Genre and 8,100
> for DGraph, to generate all the time-aware node prompts and node-aware time prompts.
> In contrast, to generate time-aware node prompts, TIGPrompt requires learning a specific prompt for each node, leading to a significantly larger number of learnable parameters: 1,609,274 for Wikipedia, 1,911,478 for Reddit, 1,250,998 for MOOC, 130,826 for Genre, and 66,610,156 for DGraph. Thus, the condition-nets in DyGPrompt is **another significant difference from previous work**, addressing **Challenge 2 in a parameter-efficient manner**.
> Hence, our approach is not just a *"more complex information fusion strategy rather than a simple concatenation"*, but using condition-net to **achieve a parameter efficient design.**
> The condition-nets also makes DyGPrompt require less training and testing times compared to TIGPrompt, as we show in our response to **W3**, further demonstrating the efficiency of using condition-net.
>
> Finally, we have updated the paper by **revising parts of the introduction** and including a detailed comparison with related work in **Appendix A.9**.

---

> ### Author Response · Authors · 2024-11-21
>
> > **W3**: Experimental setup and results
>
> **- Differences in data splits and reported results**
>
> Yes, it is indeed that *the reported results are noticeably lower than those of TIGPrompt*. This is natural as our setting focuses on a more challenging scenario with less labeled data for prompt tuning.
>
> Speficially, TIGPrompt [3] uses *"50% of the data for pre-training and 20% for prompt tuning or fine-tuning, with the remaining 30% equally divided for validation and testing."*  (see Section 4.2 of TIGPrompt). Note that pre-training data do not require any labeled examples, while prompt-tuning/fine-tuning data require labels for node classification. Hence, **TIGPrompt requires 20% labeled data for node classification**. In our experiments, we use 80% of the data for pre-training (which does not contain any labels for node classification), but **only 1% of the data serves as the training pool for prompt tuning**, with each task leveraging only 30 events (about **0.01%** of the entire dataset) for prompt tuning (where only the starting nodes in these events are labeled for node classification). Therefore, **our setting focuses on the more challenging low-resource scenario with very few labeled data**, as labeled data are generally difficult or costly to obtain in real-world applications [1,4,5]. Hence, our setting is more practical and challenging than TIGPrompt's. Additionally, we note that **TIGPrompt is a preprint paper** that has not yet been accepted and does not provide open-source code. Hence, their splits are not the sole "gold standard" to follow.
>
> Nevertheless, in the following, we **adopt the splits of TIGPrompt and conduct further experiments**, where DyGPrompt still outperforms TIGPrompt in their splits, as shown below (NC is short for node classification, LP is short for link prediction). We also updated the paper and **included these additional results into Appendix A.8**.
>
> | |Wikipedia (NC)| Wikipedia (Transductive LP) | Wikipedia (Inductive LP) |MOOC (NC)| MOOC (Transductive LP) | MOOC (Inductive LP) |
> |-|-|-|-|-|-|-|
> | TIGPrompt |78.85±1.35 | 93.95±0.47|91.35±0.38| 63.40±2.31|78.98±0.52|80.26±0.76|
> | DyGPrompt | **81.31**±1.13|**97.78**±0.36|**96.73**±0.42|**64.58**±1.95 |**87.15**±0.42|**86.14**±0.39|
>
> **- Influence of model size**
>
> As discussed in our response to **W2**, the condition-net design in DyGPrompt is more efficient than TIGPrompt, which requires learning a separate vector for each node. Hence, the prompt tuning in our DyGPrompt is more parameter efficient than that in TIGPrompt.
>
> Specifically, our dual condition-nets contains merely 3,104 learnable parameters for Wikipedia, Reddit and MOOC, 1,554 for Genre and 8,100 for DGraph, to generate all the time-aware node prompts and node-aware time prompts.
> In contrast, to generate time-aware node prompts, TIGPrompt requires learning a specific prompt for each node, leading to a significantly larger number of learnable parameters: 1,609,274 for Wikipedia, 1,911,478 for Reddit, 1,250,998 for MOOC, 130,826 for Genre, and 66,610,156 for DGraph.
> Please refer to our response to **W2** for further details.
>
> For a more comprehensive evaluation, we also conducted the following additional experiments.
>
> **- Large-scale dataset**
>
> We evaluate DyGPrompt under the same setting as introduced in Section 5.1 on a large-scale dataset DGraph [6] (3,700,550 nodes, 4,300,999 edges), designed for dynamic graph anomaly detection (a form of node classification). We report the AUC-ROC (%) in the table below, and DyGPrompt still outperforms competetive baselines, showing the robustness of DyPrompt on large-scale dataset. We also updated the paper and **include these additional results into Appendix A.6**.
>
> | | AUC-ROC (%) |
> |-|-|
> |GraphPrompt|60.18±13.06|
> |TIGPrompt|64.42±10.37|
> |DyGPrompt|**73.39**±5.85|
>
> **- Efficiency of DyGPrompt**
>
> We further compared the prompt tuning and testing times between DyGPrompt and competitive baselines for node classification tasks. As shown below (in seconds), DyGPrompt takes only marginally longer than GraphPrompt, while requring less time than TIGPrompt. The slight overhead over GraphPrompt is acceptable given the
> substantial improvement in performance. (Also note that the testing times are measured on the entire test set, which can be larger than the samples used for prompt tuning, resulting in generally longer testing times compared to prompt tuning.)
>
> We have also updated the paper and **included these results in Appendix A.7**.
>
> | | Wikipedia training | Wikipedia testing | Reddit training | Reddit testing | DGraph training | DGraph testing|
> | ------ | -------------- | ------------------- | ----------- | ---------------- |-|-|
> | GraphPrompt  | 0.147| 0.230|0.128| 0.251|0.436 | 0.324|
> | TIGPrompt    |0.312  |0.397 |0.356 |0.411|0.847 | 1.063|
> | DyGPrompt    |0.273 |0.233 |0.134 |0.274 |0.574 | 0.633|

---

> ### Author Response · Authors · 2024-11-21
>
> > **W4**: Writing
>
> Thank you for your advice. We have polished the paper, corrected typographical errors, further claimed the difference with related work, and improved Fig. 2 to enhance clarity and ease of understanding.
>
> [1] Liu et al. "Graphprompt: Unifying pre-training and downstream tasks for graph neural networks." WWW 2023.\
> [2] Sun et al. "All in one: Multi-task prompting for graph neural networks." SIGKDD 2023.\
> [3] Chen et al. "Prompt learning on temporal interaction graphs". arXiv 2024.\
> [4] Yu et al. "Generalized graph prompt: Toward a unification of pre-training and downstream tasks on graphs." TKDE 2024.\
> [5] Yu et al. "A Survey of Few-Shot Learning on Graphs: from Meta-Learning to Pre-Training and Prompt Learning". arXiv 2024.\
> [6] Huang et al. "Dgraph: A large-scale financial dataset for graph anomaly detection." NeurIPS 2022.

---

> ### Author Response · Authors · 2024-11-25
>
> As the discussion period draws to a close, we would like to confirm if our response has adequately addressed your concerns. Following your comments, we have:
>
> - Clarified the motivation of DyGPrompt.
> - Provided a detailed comparison with related work and clarified the novelty of DyGPrompt.
> - Clarified the experimental setting of DyGPrompt, compared its complexity with TIGPrompt, conducted experiments adapting the data split of TIGPrompt, compared the training and testing times of DyGPrompt with competitive baselines, and conducted experiments on a large-scale dataset.
> - Polished the paper to enhance clarity and improve readability.
>
> Thank you once again for your valuable feedback and suggestions, which have significantly helped improve our paper. We look forward to your response.

---

> ### Author Response · Authors · 2024-12-02
>
> We sincerely thank you again for reviewing our paper. With the discussion period nearing its end, we would like to confirm whether our responses have adequately addressed your concerns. Please feel free to reach out if you have any additional questions. We are more than happy to provide further explanations.

---

> > ### Comment · Reviewer_BsEP · 2024-12-02
> >
> > Thank you for your efforts. This version shows significant improvement, and I’ve increased my score accordingly.

---

### Official Review · Reviewer_y8dA · 2024-11-06

**Soundness:** 3
**Presentation:** 3
**Contribution:** 3
**Rating:** 6
**Confidence:** 4

**Summary:**

This paper studies the problem of adapting pre-trained dynamic GNNs. The authors propose DYGPROMPT, a dual prompt mechanism with node and time prompts, supported by dual condition-nets, to bridge the gaps between pre-training and downstream objectives as well as temporal inconsistencies, achieving improved adaptability and performance across dynamic graph tasks.

**Strengths:**

S1. I like the idea of using dual prompt mechanism to bridge task and temporal gaps.

S2. Experiments show DYGPROMPT is effective on small real-world graphs.

**Weaknesses:**

W1. The introduction of dual prompts and dual condition networks enhances adaptability but significantly increases model complexity. I would suggest the authors to add some analysis or experiments to demonstrate DYGPROMPT’s performance in terms of computational efficiency.

W2. I have concerns about the experimental setup. Using the first 80% of interactions for pre-training is uncommon in pre-trained frameworks. For example, TIGPROMPT uses only 20% of the data, yielding surprisingly strong results in comparison.

W3. The discussion of current dynamic graph prompt learning methods, such as TIGPROMPT, is brief and lacks an in-depth analysis of their limitations or how DYGPROMPT specifically addresses these issues. Additionally, Section 4 contains extensive information on previous works, making it difficult to discern the paper’s core contributions.

**Questions:**

Q1. In previous work, prompts are often implemented as vector concatenations. What are the author’s considerations for using dot-product instead?

Q2. Personally, I am curious whether a pre-trained time encoder can address questions such as “What is the time of the next interaction?”

Q3. How does DYGPROMPT perform on large-scale dynamic graphs?

---

> ### Author Response · Authors · 2024-11-21
>
> We sincerely appreciate your insightful feedback and constructive comments, which have greatly contributed to refining our work. Our responses to your concerns are as follows, and we are eager to discuss any further questions you may have.
>
> > **W1: Complexity of DyGPrompt**
>
> We agree that integrating the dual condition-net increases the complexity of DyGPrompt compared to standard prompt tuning methods [1,2,3]. However, as analyzed in the **Complexity Analysis in Appendix A.2**, the complexity of DyGPrompt is primarily dominated by the pre-trained dynamic graph encoder, similar to how standard prompt tuning methods are also dominated by the pre-trained graph encoder [3]. The addition of the dual condition-net results in only a relatively small increase in the complexity of prompt tuning, while bringing substantial performance improvement.
>
> Moreover, we compared the prompt tuning and testing times between DyGPrompt and competitive baselines for node classification tasks. We also included a large-scale dataset, DGraph [10] with 3,700,550 nodes and 4,300,999 edges (also see our response to **Q3**), for a more comprehensive comparison. As shown below (time measured in seconds), DyGPrompt takes only marginally longer than GraphPrompt, while requring less time than TIGPrompt. The slight overhead over GraphPrompt is acceptable given the
> substantial improvement in performance. (Also note that the testing times are measured on the entire test set, which can be larger than the samples used for prompt tuning, resulting in generally longer testing times compared to prompt tuning.)
>
> We have also updated the paper and **included these results in Appendix A.7**.
>
> | | Wikipedia training | Wikipedia testing | Reddit training | Reddit testing | DGraph training | DGraph testing|
> | ------ | -------------- | ------------------- | ----------- | ---------------- |-|-|
> | GraphPrompt  | 0.147| 0.230|0.128| 0.251|0.436 | 0.324|
> | TIGPrompt    |0.312  |0.397 |0.356 |0.411|0.847 | 1.063|
> | DyGPrompt    |0.273 |0.233 |0.134 |0.274 |0.574 | 0.633|
>
>
> > **W2: Experimental setup**
>
> TIGPrompt [4] uses *"50% of the data for pre-training and 20% for prompt tuning or fine-tuning, with the remaining 30% equally divided for validation and testing."*  (see Section 4.2 of TIGPrompt). Note that pre-training data do not require any labeled examples, while prompt-tuning/fine-tuning data require labels for node classification. Hence, **TIGPrompt requires 20% labeled data for node classification**. In our experiments, we use 80% of the data for pre-training (which does not contain any labels for node classification), but **only 1% of the data serves as the training pool for prompt tuning**, with each task leveraging only 30 events (about **0.01%** of the entire dataset) for prompt tuning (where only the starting nodes in these events are labeled for node classification). Therefore, **our setting focuses on the more challenging low-resource scenario with very few labeled data**, as labeled data are generally difficult or costly to obtain in real-world applications [1,3,5]. Hence, our setting is more practical and challenging than TIGPrompt's. Additionally, we note that **TIGPrompt is a preprint paper** that has not yet been accepted and does not provide open-source code. Hence, their splits are not the sole "gold standard" to follow.
>
> Naturally, using a larger amount of labeled data for prompt tuning would significantly boost model performance, which is why our reported results for TIGPrompt are noticeably lower than those in their paper. To further showcase that our approach is robust across different experimental settings, in the following, we **adopt the splits of TIGPrompt and conduct further experiments**, where DyGPrompt still outperforms TIGPrompt in their splits, as shown below (NC is short for node classification, LP is short for link prediction). We also updated the paper and **included these additional results into Appendix A.8**.
>
> | |Wikipedia (NC)| Wikipedia (Transductive LP) | Wikipedia (Inductive LP) |MOOC (NC)| MOOC (Transductive LP) | MOOC (Inductive LP)|
> |-|-|-|-|-|-|-|
> | TIGPrompt |78.85±1.35 | 93.95±0.47|91.35±0.38| 63.40±2.31|78.98±0.52|80.26±0.76|
> | DyGPrompt | **81.31**±1.13|**97.78**±0.36|**96.73**±0.42|**64.58**±1.95 |**87.15**±0.42|**86.14**±0.39|

---

> ### Author Response · Authors · 2024-11-21
>
> > **W3: Discussion of related work and contribution of DyGPrompt**
>
> We summarize the comparison of DyGPrompt with representative graph prompt learning methods, including GraphPrompt [1] and ProG [2] for static graphs, and the contemporary work TIGPrompt [4] for dynamic graphs (using its best-performing variant, Projection TProG, for illustration), as below.
>
> | |Explicit node prompt|Explicit time prompt|Time-aware node prompts|Node-aware time prompts|Condition-net|
> |-|-|-|-|-|-|
> |GraphPrompt|$\checkmark$|$\times$|$\times$|$\times$|$\times$|
> |ProG|$\checkmark$|$\times$|$\times$|$\times$|$\times$|
> |TIGPrompt|$\times$|$\times$ |$\checkmark$|$\times$ |$\times$|
> |DyGPrompt| $\checkmark$|$\checkmark$ | $\checkmark$| $\checkmark$|$\checkmark$|
>
> More specifically, first, a *node prompt* modifies the node features to reformulate the task input and bridge the task gap (Eq. 4), while a *time prompt* adjusts the time features to capture the temporal evolution of the dynamic graph (Eq. 5). The dual prompts are motivated by **Challenge 1 in Section 1**. Previous static graph prompt learning methods (GraphPrompt and ProG) only utilize a node prompt, neglecting the temporal gap between pre-training and downstream tasks. Furthermore, TIGPrompt, falls short of explicitly addressing each gap individually.
> The dual prompts also demonstrate empirical benefits. As shown in the ablation study in Table 2, Variant 2 (using only node prompt) and Variant 3 (using only time prompt) consistently outperform Variant 1. Additionally, Variant 4 (incorporating both node and time prompts) generally outperforms Variants 2 and 3, demonstrating the effectiveness of node and time prompts in bridging the task gap and temporal gap between pre-training and downstream tasks.
>
> Second, *time-aware node prompts* adjust node prompts, and ultimately node features, to reflect temporal influence on the node prompts (Eq. 7) instead of using a fixed node prompt across time. On the other hand, *node-aware time prompts* adjust the time prompt for each node by incorporating node-specific characteristics (Eq. 9) instead of using a fixed time prompt across nodes. Consequently, node-time patterns and their dynamic interplay are captured, addressing **Challenge 2 in Section 1**. Static graph prompt learning methods lack such designs, failing to capture the mutual characterization between node and time patterns. Moreover, TIGPrompt considers only the temporal impact on nodes by generating time-aware node prompts, but neglects that time prompts for each node can also be influenced by node features, hindering its ability to model the scenario where different nodes may exhibit divergent behaviors even at the same time point (as illustrated in Fig. 1).
> As shown in the ablation study in Table 2, Variant 5 (with time-aware node prompts) outperforms Variant 2 (without time-aware node prompts), Variant 6 (with node-aware time prompts) outperforms Variant 3 (without node-aware time prompts), and DyGPrompt (with both) outperforms Variant 4 (without either), further demonstrating the effectiveness of our designs.
>
> Third, we propose *dual condition-nets* to effectively generate time-aware node prompts (Eq. 6) and node-aware time prompts (Eq. 8) while avoid overfitting in a parameter-efficient way. These condition-nets generate prompts conditioned on the input features rather than directly parameterizing the prompts, significantly reducing the number of learnable parameters in the downstream prompt-tuning phase.
> In our implementation, we use a 2-layer MLP as the condition-net. Specifically, our dual condition-nets contains merely 3,104 learnable parameters for Wikipedia, Reddit and MOOC, 1,554 for Genre and 8,100
> for DGraph, to generate all the time-aware node prompts and node-aware time prompts.
> In contrast, to generate time-aware node prompts, TIGPrompt requires learning a specific prompt for each node, leading to a significantly larger number of learnable parameters: 1,609,274 for Wikipedia, 1,911,478 for Reddit, 1,250,998 for MOOC, 130,826 for Genre, and 66,610,156 for DGraph. Moreover, as shown in our response to **W1**, DyGPrompt requires less training and testing time compared to TIGPrompt, further demonstrating the efficiency of using condition-net. Thus, the condition-nets in DyGPrompt is **another significant difference from previous work**, addressing **Challenge 2 in a parameter-efficient manner**.
>
> Finally, we have updated the paper by **revising parts of the introduction** and including a detailed comparison with related work in **Appendix A.9**. We have also more explicitly differentiated our proposal versus previous works in Section 4.

---

> ### Author Response · Authors · 2024-11-21
>
> > **Q1**: Element-wise multiplication for modification
>
> We use element-wise multiplication, in line with previous graph prompt learning methods on static graphs [1,3,8,9,11]. We have updated the paper and clarified this in Section 4.3.
>
> > **Q2**: Prediction of time of the next interaction
>
> This is a great question. In our current temporal link prediction task, we predict if there is an interaction at a specific time. A simple workaround is to probe various time points at reasonable increments, and predict if there is an interaction at those time points, thereby estimating the time point when the next interaction will occur. Of course, this probing might be inefficient. It would be interesting to explore a more efficient and direct solution in future work.
>
> >**Q3**: Large-scale dataset.
>
> We evaluate DyGPrompt under the same setting as introduced in Section 5.1 on a large-scale dataset DGraph [10] (3,700,550 nodes, 4,300,999 edges), designed for dynamic graph anomaly detection (a form of node classification). We report the AUC-ROC (%) in the table below, and DyGPrompt still outperforms competetive baselines, showing the robustness of DyPrompt on large-scale dataset. We also updated the paper and **include these additional results into Appendix A.6**.
>
>
> | | AUC-ROC (%) |
> |-|-|
> |GraphPrompt|60.18±13.06|
> |TIGPrompt|64.42±10.37|
> |DyGPrompt|**73.39**±5.85|
>
> [1] Liu et al. "Graphprompt: Unifying pre-training and downstream tasks for graph neural networks." WWW 2023.\
> [2] Sun et al. "All in one: Multi-task prompting for graph neural networks." SIGKDD 2023.\
> [3] Yu et al. "Generalized graph prompt: Toward a unification of pre-training and downstream tasks on graphs." TKDE 2024.\
> [4] Chen et al. "Prompt learning on temporal interaction graphs". arXiv 2024.\
> [5] Yu et al. "A Survey of Few-Shot Learning on Graphs: from Meta-Learning to Pre-Training and Prompt Learning". arXiv 2024.\
> [6] Xu et al. "Inductive Representation Learning on Temporal Graphs". ICLR 2020.\
> [7] Rossi et al. "Temporal graph networks for deep learning on dynamic graphs". arXiv 2020.\
> [8] Yu et al. "MultiGPrompt for Multi-Task Pre-Training and Prompting on Graphs". WWW 2024.\
> [9] Yu et al. "HGPrompt: Bridging Homogeneous and Heterogeneous Graphs for Few-shot Prompt Learning." AAAI 2024.\
> [10] Huang et al. "Dgraph: A large-scale financial dataset for graph anomaly detection." NeurIPS 2022.

---

> ### Author Response · Authors · 2024-11-25
>
> As the discussion period is coming to an end, we would like to check whether our responses have addressed your questions. Following your comments, we:
>
> - Conducted experiments to compare the training and testing times of DyGPrompt with competitive baselines.
> - Clarified the experimental setting and performed experiments using the data split of TIGPrompt.
> - Provided a detailed comparison with related work.
> - Clarified the use of element-wise multiplication for feature modification.
> - Discuss the prediction of time of the next interaction.
> - Conducted experiments on a large-scale dataset to address your concerns.
>
> Thank you again for your valuable comments and suggestions to improve our paper. We look forward to your reply.

---

> ### Author Response · Authors · 2024-12-02
>
> Thank you once again for taking the time to review our paper. As the discussion period is approaching its end, we are keen to know whether our responses have sufficiently addressed your concerns. Moreover, we are more than willing to provide additional explanations if you have any further questions.

---

> ### Comment · Reviewer_y8dA · 2024-12-02
>
> Thank you for your detailed responses. After reviewing your replies and the revised paper, I have no further questions and am willing to update my score.

---

### Author Response · Authors · 2024-11-21
**General response**

We sincerely appreciate the time and effort you have dedicated to reviewing our paper. Your insights are invaluable in refining our work. We are inspired by the positive feedback received, and we would like to highlight a few key areas:

- **Novelty/contribution of prompt tuning for dynamic graphs**: We are deeply grateful for the reviews describing our work as "novel" (Reviewer ngqA, obkg), "reasonable" (Reviewer ngqA), "creative" (Reviewer obkg), "like the idea" (Reviewer y8dA).
- **Extensive Experiments**: The consensus among multiple reviewers (Reviewers y8dA, ngqA, obkg) regarding the extensiveness of our experiments further validates our approach.
- **Good presentation**: We are pleased that the presentation of our work were well-received, as indicated by comments like "well-written and easy to follow" (Reviewers ngqA, obkg).

In response to your comments, we have included the following as our rebuttal:
- Additional experiments on a large-scale dataset.
- Additional experiments to compare prompt tuning and testing times.
- A detailed comparison with related graph prompt learning methods to highlight our contributions.
- Clarification of our experimental settings, and additional experiments on a different data split.
- Addressing all other minor issues, such as technical clarifications and corrections of typographical errors.

We have also updated the paper accordingly, with changes marked in blue.

---

### Meta-Review · Area_Chair_cw7z · 2024-12-21

**Metareview:**

The paper proposes DYGPROMPT, a dual prompt mechanism with node and time prompts for pre-training dynamic GNNs. The problem is interesting and popular. The proposed model is reasonable. The experiment results show that the proposed method outperforms baseline methods. The paper writing is good. Several issues were raised by reviewers to improve the paper, such as a lack of discussion of related works (e.g., TIGPrompt), relatively small datasets, and complexity analysis. Reviewers are relatively positive about this work.

**Additional Comments On Reviewer Discussion:**

I posted a discussion, but no one replied. They are relatively positive about this work, according to reviews.

---

### Decision · Program_Chairs · 2025-01-22

Accept (Poster)